# Past Ocean surface density from planktonic foraminifera calcite $\delta^{18}$O

Thibaut Caley[1], Niclas Rieger[2,3], Martin Werner[4], Claire Waelbroeck[5], Héloise Barathieu[1], Tamara Happé[6], Didier M. Roche[7,8]

1) Univ. Bordeaux, CNRS, Bordeaux INP, EPOC, UMR 5805, 33600 Pessac, France

2) Institut de Ciències del Mar (ICM) - CSIC, Pg. Marítim de la Barceloneta 37, Barcelona, Spain

3) Centre de Recerca Matemàtica (CRM), Departament de Física, Bellaterra, Spain

4) Alfred Wegener Institute Helmholtz Centre for Polar and Marine Research, Bremerhaven, Germany

5) LOCEAN-IPSL, CNRS, Sorbonne Université, Paris, France

6) Institute for Environmental Studies, Vrije Universiteit Amsterdam, Amsterdam, The Netherlands

7) Laboratoire des Sciences du Climat et de l'Environnement, LSCE/IPSL, CEA-CNRS-UVSQ, Université Paris-Saclay, 91191 Gif-sur-Yvette, France

8) Earth and Climate Cluster, Faculty of Earth and Life Sciences, Vrije Universiteit Amsterdam, Amsterdam, the Netherlands

Correspondence to: Thibaut Caley (thibaut.caley@u-bordeaux.fr)

**Abstract**
Density of seawater is a critical property that controls ocean dynamics. Previous works
suggest the use of the $\delta^{18}O$ calcite of foraminifera as a potential proxy for paleodensity.
However, potential quantitative reconstructions were limited to the tropical and subtropical
surface ocean and without an explicit estimate of the uncertainty in calibration model
parameters. We developed the use of the $\delta^{18}Oc$ of planktonic foraminifera as a surface
paleodensity proxy using Bayesian regression models calibrated to annual surface density.
Predictive performance of the models improves when we account for inter-species specific
differences.
We investigate the additional uncertainties that could be introduced by potential evolution
of the $\delta^{18}Oc$-density relationship with time (from the last glacial maximum (LGM) to the
preindustrial (PI)) through the combination of past isotope enabled climate model
simulations and a foraminiferal growth module. We demonstrate that additional
uncertainties are weak globally, except for the Nordic Seas region.
We applied our Bayesian regression model to LGM and Late Holocene (LH) $\delta^{18}Oc$
foraminifera databases to reconstruct annual surface density during these periods. We
observe stronger LGM density value changes at low latitudes compared to mid latitudes.
These results will be used to evaluate numerical climate models in their ability to simulate
ocean surface density during the extreme climatic period of the LGM.
The new calibration has great potential to reconstruct the past temporal evolution of ocean
surface density over the Quaternary. Under climates outside the Quaternary period and in
ocean basins characterized by anti-estuary circulation, like the current Mediterranean Sea
and Red Sea, our calibration could provide density estimates with larger uncertainty, a point
that requires further investigations.












1. Introduction

Temperature and salinity control the density of seawater and therefore the ocean dynamics too. Reconstruction of past ocean surface temperature with reasonable uncertainties is possible (MARGO, 2009; Tierney et al., 2020b) but reconstructions of past surface salinity remain very challenging in paleoceanography. When the current uncertainties on past temperature and salinity reconstructions are cumulated, it becomes unreasonable to combine these two parameters in order to quantify past ocean density and dynamics (Schmidt, 1999).

Rather than using the combination of temperature and salinity, previous works suggest the use of the $\delta^{18}O$ of foraminiferal calcite as a potential proxy for paleodensity (Lynch-Stieglitz et al., 1999; Billups and Schrag, 2000, LeGrande et al., 2004; Lynch-Stieglitz et al., 2007). The oxygen isotopic composition of foraminifera calcite is controlled by 1) the temperature dependence of the equilibrium fractionation during calcite precipitation and 2) the isotopic composition of seawater in which the shell grows (Urey, 1947; Shackleton, 1974). Except in areas of sea ice formation or melt, the isotopic composition of seawater ($\delta^{18}O_{sw}$) is regionally related to salinity, since they are affected by processes such as evaporation, precipitation, and the water masses advection and mixing (Craig and Gordon, 1965). Therefore, both temperature and $\delta^{18}O_{sw}$ changes that affect the foraminifera $\delta^{18}O$ calcite ($\delta^{18}Oc$) signal are also the processes that ultimately define the seawater density in which the foraminifera calcifies (Lynch-Stieglitz, 1999; Billups and Schrag, 2000).

In addition to temperature and $\delta^{18}Osw$, the shell $\delta^{18}Oc$ signal can also be potentially influenced by biological processes, such as: 1) photosynthesis in algal symbionts (Duplessy et al., 1970; Ravelo and Fairbanks, 1992; Spero and Lea, 1993; Spero et al., 1997) and biases due to the formation of gametogenic or ontogenetic calcite (Williams et al., 1979; Spero and Lea, 1996; Hamilton et al., 2008), 2) changes in pH and carbonate ion concentration [$CO_3^{2-}$] (Spero et al., 1997; Bijma et al., 1999; Zeebe, 1999), 3) dissolution and recrystallization for shells deposited in bottom waters undersaturated in [$CO_3^{2-}$] (Schrag et al., 1995), and 4) bioturbation (Waelbroeck et al., 2005). The four processes mentioned above have not been clearly demonstrated. In addition, the carbonate ion effect has been shown to have no detectable influence (Köhler and Mulitza, 2024) and core top data have been selected to limit the bioturbation effect (Waelbroeck et al., 2005). Therefore, we do not account for these processes. Transport of foraminifera shells by currents is another process that could lead to discrepancies between recorded $\delta^{18}Oc$ and calculated $\delta^{18}Oc$ or hydrographic data. However, this effect is likely minimal because the ambient water mass is transported together with the shells. Later in this study (Sect. 3.1.2), we confirm that planktonic foraminifera $\delta^{18}Oc$ is mainly related to the surface ocean density, growth season and habitat depth, with weak additional influence from biological processes.

Previously, Billups and Schrag (2000) used $\delta^{18}Oc$ from the mixed layer planktonic foraminifera (*Globigerinoides ruber* and *Trilobatus sacculifer*) as a proxy of surface water density. They limited their study to the tropical and subtropical surface ocean.

In this study we investigate the use of planktonic foraminifera δ¹⁸Oc as a surface
paleodensity proxy for the whole ocean, from low to high latitudes, using various
foraminifera species: *Globigerinoides ruber* (*G. ruber*), *Trilobatus sacculifer* (*T. sacculifer*),
*Globigerina bulloides* (*G. bulloides*), *Neogloboquadrina incompta* (*N. incompta*), and
*Neogloboquadrina pachyderma* (*N. pachyderma*). Compared to Billups and Schrag (2000), we
use extended late Holocene (LH) and last glacial maximum (LGM) δ¹⁸Oc databases
(Malevitch et al., 2019; Caley et al., 2014, Waelbroeck et al., 2014; Tierney et al., 2020b). We
develop mean annual surface density calibration models using a Bayesian approach. We also
use numerical climate simulations obtained with isotope enabled climate models
(iLOVECLIM and ECHAM5/MPI-OM) and a foraminiferal growth module (FAME) (Roche et al.,
2018) to investigate the specific seasonal dynamic and depth habitat preference of
foraminifera (Roche et al., 2018; Schiebel and Hemleben 2018). We discuss the applicability
and validity of the foraminifera δ¹⁸Oc to the past quantification of surface ocean density. We
then reconstruct past surface density changes during the LGM.

2.  Methods
2.1 Planktonic foraminifera δ¹⁸O databases
We compiled global foraminifera oxygen isotopic datasets from published LH and LGM
measurements to allow reconstruction of past density. We used core-top and LH records of
planktonic foraminifera δ¹⁸Oc from Malevich et al. 2019 dataset that include records from
the Multiproxy Approach for the Reconstruction of the Glacial Ocean (MARGO) (Waelbroeck
et al., 2005) with additional sources. This dataset consists of 2,636 observations with 1,002
for *G. ruber*, 635 for *G. bulloides*, 442 for *T. sacculifer*, 132 for *N. incompta* and 425 for *N.*
*pachyderma* (Malevich et al., 2019). Similarly to Malevich et al. 2019, we gridded the core-
top data to reduce the impact of spatial clustering by averaging samples for each species to
the nearest 1°X 1° grid point. So doing, we obtained a total of 1,415 grid points.
For the LGM time period, records derived in part from the MARGO collection (Waelbroeck et
al., 2014), with additional data from Caley et al., 2014, Tierney et al., 2020b, and from more
recent studies (34 measurements). The final dataset consists of 474 observations.
Chronostratigraphic quality for the LGM and LH is consistent between all the published
databases, the additional observations and use the same MARGO definition (MARGO, 2009).
2.2 Ocean dataset
In order to establish and test our calibrations between foraminifera δ¹⁸Oc and observed
surface density, we used different ocean datasets. We used the Multi Observation Global
Ocean Sea Surface density product for our core-top and Late Holocene calibration models
(Droghei et al., 2016; 2018). This means that we calibrated Late Holocene core-top samples
against observed density fields influenced by anthropogenic climate change, an issue that
affects all core-top calibrations. To test the residual of our models against sea surface
temperature and salinity (SST and SSS respectively) we used WOA18 products (Locarnini et
al., 2018; Zweng et al., 2018).

## 2.3 Bayesian calibration models and evaluation


Following the general approach of Malevich et al. (2019), we use Bayesian regressions to model the relationship between the calcite oxygen isotopic composition of planktonic foraminifera, $\delta^{18}Oc$ (‰ VPDB), and annual mean surface density, $\rho$ (kg/m$^3$ relative to the water density of 1000 kg/m$^3$). By explicitly estimating uncertainty in the calibration model parameters, each model produces a full posterior predictive distribution for the predictant $\rho$. We implement three Bayesian models—two pooling models with first- and second-degree polynomials, and a hierarchical first-degree polynomial model—using Markov chain Monte Carlo (MCMC) methods (see Kruschke, 2014; McElreath, 2018 for review).

### 2.3.1 Three Bayesian calibration models

1.  **First-Degree Polynomial (Pooled), poly1_pool:**
    A simple linear regression is fit to all foraminifera species combined:
    $$\rho \sim N(\mu, \sigma^2), \quad \mu = \beta_0 + \beta_1 \delta^{18}O_c.$$
    Weakly informative data-adaptiv normal hyperpriors are used for $\beta_0$ and $\beta_1$, and an exponential prior for the noise term sigma. This pooled model assumes a common relationship across all foraminifera species (see Appendix).
2.  **Second-Degree Polynomial (Pooled), poly2_pool:**
    Motivated by empirical evidence (e.g., Billups and Schrag (2000)), the second model incorporates a quadratic term:
    $$\rho \sim N(\mu, \sigma^2), \quad \mu = \beta_0 + \beta_1 \delta^{18}O_c + \beta_2 (\delta^{18}O_c)^2.$$
    Again, we apply weakly informative normal priors for the $\beta_i$ parameters, ensuring flexibility while constraining the plausible range based on the observed data.
3.  **First-Degree Polynomial (Hierarchical), poly1_hier:**
    The third model recognizes that species-specific differences in calcification, depth, seasonality and vital effects can affect $\delta^{18}O_c$ (Malevich et al., 2019). Hence, we use a hierarchical structure:
    $$\rho \sim N(\mu_s, \sigma_s{}^2), \quad \mu_s = \beta_{s,0} + \beta_{s,1} \delta^{18}O_c.$$
    where each species s has its own intercept ($\beta_{s,0}$) and slope ($\beta_{s,1}$). These species-level parameters are drawn from common hyperdistributions $v_i$ and $\kappa_i$ (Appendix A), ensuring partial pooling of information across species.

### 2.3.2 Model fitting and evaluation

182

All models were fitted with six independent MCMC chains of 4000 iterations each, discarding the first 2000 as burn-in. We used rank-normalized $\hat{R}$ (Vehtari et al. 2021) to assess convergence, finding all values below 1.05. Prior and posterior predictive checks confirmed the adequacy of the models. To compare predictive performance, we computed the expected log pointwise predictive density (ELPD) via Pareto-smoothed importance sampling leave-one-out cross-validation (LOO) (Vehtari et al., 2017), which provides a principled basis for selecting the model that best characterizes the relationship between $\delta^{18}Oc$ and $\rho$. The

ELPD measures the expected predictive accuracy of a Bayesian model. It is defined as the
sum over all data points of the expected log posterior predictive density (Gelman et al.,
2014). In our case, a higher ELPD means the model makes sharper and more accurate
density predictions.

2.4 Isotope enabled numerical climate models
2.4.1 The iLOVECLIM model
The iLOVECLIM (version 1.1.3) earth system model of intermediate-complexity is a derivative
of the LOVECLIM-1.2 climate model extensively described in Goosse et al. (2010). From the
original model, we retain the atmospheric (ECBilt, resolution of 5.6° in latitude and
longitude), oceanic (CLIO, 3x3° horizontal resolution, 20 vertical layers and a free surface),
vegetation (VECODE) and land surface (LBM) components and develop a complete,
conservative, water isotope cycle through all cited components. A detailed description of the
method used to compute the oxygen isotopes in iLOVECLIM can be found in Roche (2013)
and the validation of model results can be found in Roche and Caley (2013), Caley and Roche
(2013) and Extier et al., 2024.
We use the boundary conditions defined in/by the PMIP2 protocol to simulate the annual
LGM climate (Caley et al., 2014). Details about the model simulations (LGM and pre-
industrial (PI)) and validation of results for oxygen stable isotopes and temperature can be
found in Caley et al. 2014.
2.4.2 The ECHAM5/MPI-OM model
We also use the ECHAM5/MPI-OM coupled General Circulation Model (GCM), also
previously named community Earth system model COSMOS. It is a fully coupled ocean–
atmosphere–sea ice– land surface model (Jungclaus et al., 2006) with stable water isotope
diagnostics in all relevant model components. Mass, energy, and momentum fluxes, as well
as the related isotope masses of $H_2^{18}O$ and HDO, are exchanged between the atmosphere
and ocean once per day. Further details about the model can be found in Werner et al.,
217 2016.

We used monthly outputs of the two simulations performed for the PI and for the LGM
climate as described and evaluated for oxygen stable isotopes in Werner et al., 2016.

2.5  The FAME module
Foraminifera as Modelled Entities (FAME; Roche et al., 2018) is a foraminiferal growth
module that tackles the dynamic seasonal and depth habitat of planktonic foraminifera. The
module predicts the presence or absence of commonly used planktonic foraminifera and
their $\delta^{18}O$ values. It uses a very limited number of parameters, almost all derived from
culture experiments (Lombard et al., 2009).

3.    Results and discussion
3.1 Ocean surface density from planktonic foraminifera calcite $\delta^{18}$O
The three Bayesian calibration models reasonably replicate core top data spread when we
predict surface density (Fig. 1).

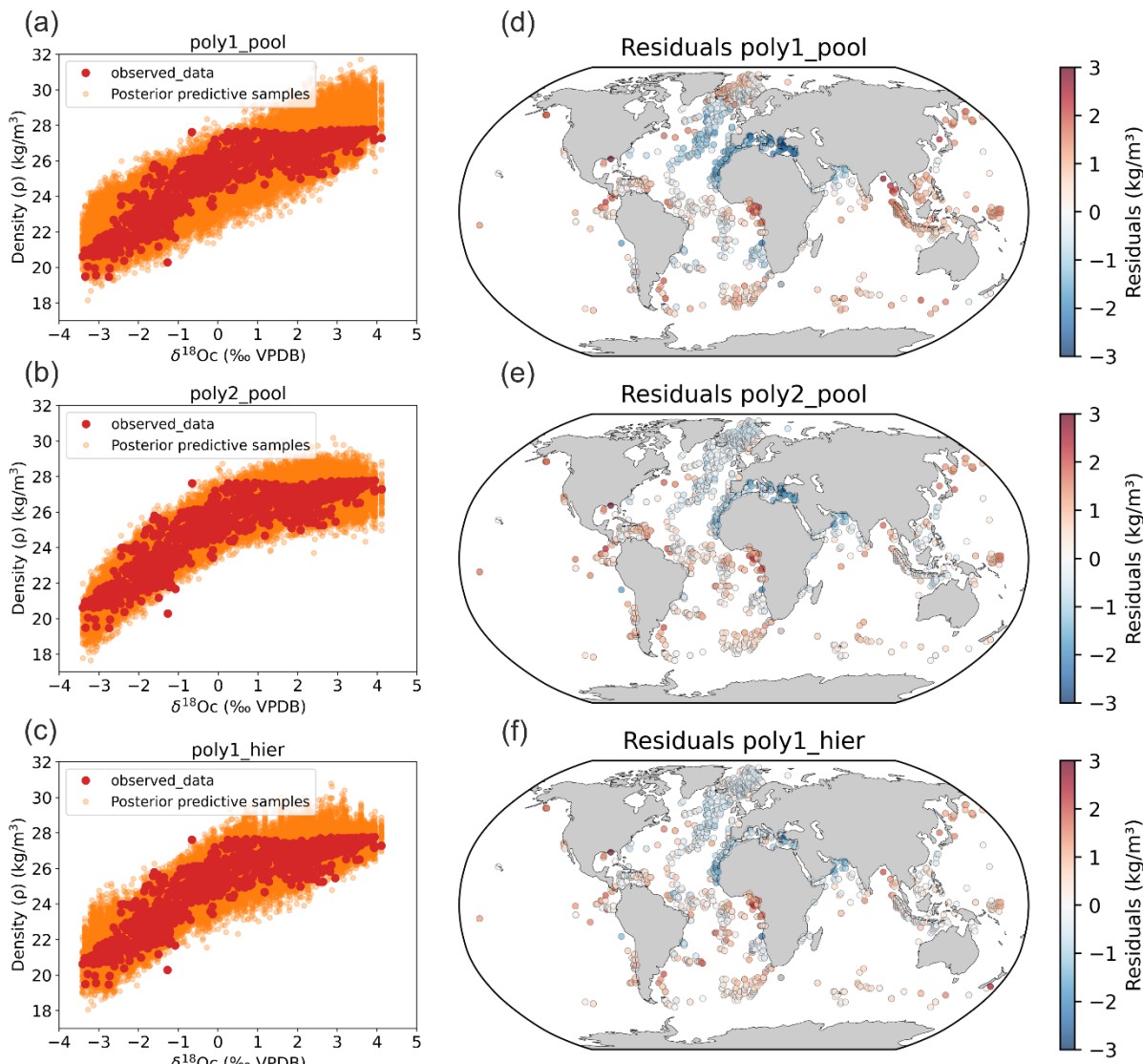


Figure 1: Bayesian calibration models for late Holocene core-top samples against observed
density. (a), (b) and (c) The three Bayesian regression models between foraminifera $\delta^{18}$Oc
and annual surface density and (d), (e) and (f) associated density residuals (predicted -
observed).
Compared to the Billups and Schrag (2000) study which was restricted to the 21-26 density
range in tropical and subtropical regions, our models provide estimates of the density
changes over the whole density range from 19 to 28 (Fig. 1).  In our new calibrations, we also
explicitly estimate the uncertainty in calibration model parameters (Fig. 1) using a Bayesian
approach to calculate robust confidence intervals.
We observe a saturation of density values close to 28 in the calibrations that correspond to
high latitudes regions (Nordic Seas and Austral Ocean). When density is already high,
temperature changes have a smaller effect. Cold water is already dense, so cooling it further
doesn't increase density as much. Consequently, we observe a sensitivity decrease. The rate
of change of density with respect to temperature flattens out, meaning that the system
becomes less responsive to temperature changes. Small changes in temperature and salinity
no longer cause significant shifts in density. This behavior reflects to the non-linearity of the
seawater equation of state.  Although the regression becomes less predictive in this range,
the estimated density values remain correct and are not expected to change strongly as
ocean surface density approaches its upper limits.
3.1.1 Model comparison and residuals
Looking at the density residual (predicted - observed) for the three models, the first model
(linear pools) has the highest values of residual and the third model (hierarchical design)
performs best (Fig. 1). The second model performs clearly better than the first one but less
than the hierarchical design. This is supported by model evaluation using log pointwise
predictive density (ELPD) (Vehtari et al., 2017) (Fig. 2). Predictive performance of the model
improves when we account for species-specific differences and species-specific prediction
uncertainty (sigma) in surface density predictions vary between foraminifera species (Fig. 2).

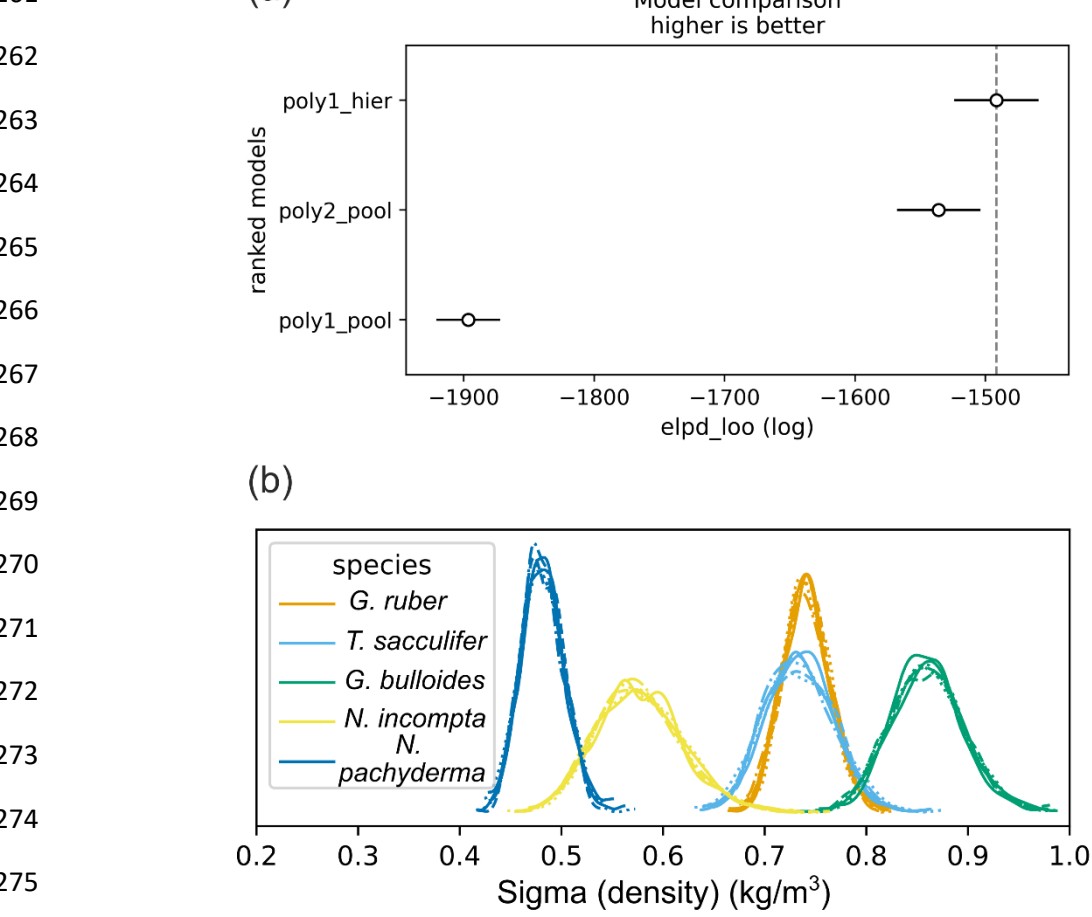

Figure 2: Model comparison and prediction uncertainty across species. (a) Expected log pointwise predictive density (ELPD) for the three models; higher values indicate better predictive performance. (b) Posterior distributions of the prediction-error parameter (sigma density) from the hierarchical model for each foraminifera species (six MCMC chains shown). Among these, *N. pachyderma* exhibits the lowest uncertainty, while *G. bulloides* shows the highest.

We still observe residuals with the hierarchical model (Fig. 1), so we checked their relation to SST and SSS (Fig. 3). The residuals of the pooled linear annual calibration model exhibit a relationship with SST and a linear relationship with SSS with a relatively high correlation ($R^2$ = 0.55, p-value <0.05). In contrast, the residuals of the hierarchical annual calibration model show no correlation to SST ($R^2$ = 0, p-value <0.05) and only a very weak correlation with SSS ($R^2$ = 0.21, p-value <0.05).  This suggests that factors other than SST and SSS influence the remaining residual structures, and some may be indirectly associated with SSS gradients. Indeed, ecological factors (e.g. seasonality and habitat depth) and secondary environmental parameters (e.g. nutrients and light penetration) may also contribute. This is supported by the fact that the residual of individual species (Fig.  3) show various significant relations (p-value <0.05) with SSS, with $R^2$ values of 0.17 for *G. ruber*, 0.12 for *T. sacculifer*, 0.54 for *G. bulloides*, 0.15 for *N. incompta*, and 0.32 for *N. pachyderma*. For example, negative residuals are observed in the Benguela, Canary, Peru and North Arabian regions (Fig. 1). All these coastal areas correspond to upwelling systems and previous work already suggested that foraminifera species could have a preference for nutrient-rich waters with high turbidity. This is particularly true for the seasonal specie *G. bulloides* (Peeters et al., 2002; Gibson et al., 2016). The $\delta^{18}Oc$ may therefore be biased toward colder temperatures even when accounting for seasonality and species-specific sensitivity (Malevich et al., 2019).  This could explain why all three models yield lower densities than the observed annual mean densities in the upwelling zones. The negative density residuals in these upwelling regions may reflect this habitat preference (Fig. 1).

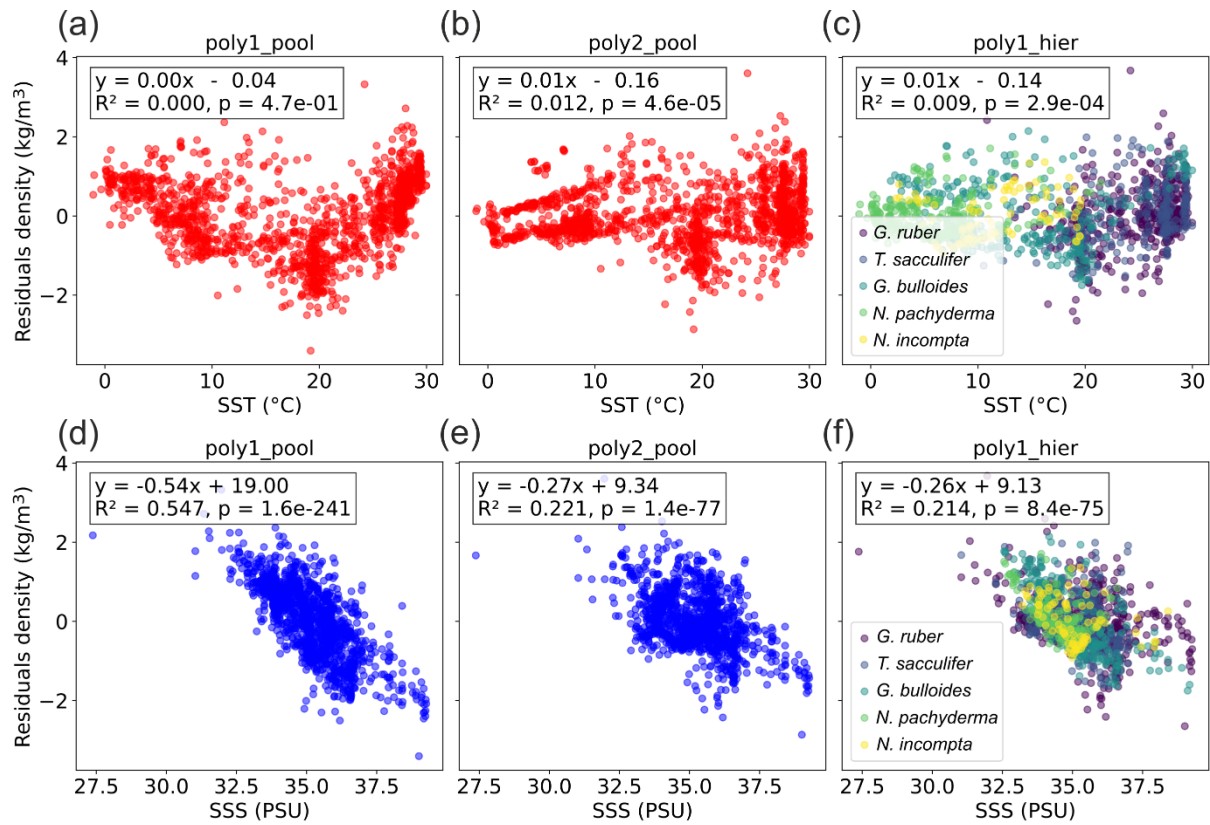

Figure 3: Relation between density residuals (predicted - observed) and (a), (b) and (c) for SST and (d), (e) and (f) for SSS (WOA18 products, Locarnini et al., 2018; Zweng et al., 2018) for the three Bayesian regression models. $R^2$ and p-values are indicated.

Strong negative residuals are also observed in the eastern part of the Mediterranean Sea. Malevich et al. 2019 reported reduced performance of their hierarchical seasonal calibration model for $\delta^{18}Oc$ and SST in this region and attributed it to the unusual behavior of *G. ruber,* potentially linked to depth-habitat migration. But estimation of seasonality for this region could also be problematic and play a role as highlighted in the study of Ayache et al. 2024. Alternatively, biases in Mediterranean net freshwater fluxes and thermohaline circulation could affect late Holocene $\delta^{18}Oc$ values (Ayache et al., 2014). Future modelling developments, such as the use of high-resolution regional model in combination with the FAME module, could help to better understand the relation between $\delta^{18}Oc$, density, temperature and $\delta^{18}Osw$ during past climate changes in the Mediterranean Sea.

We also observe high positive residual values in the Equatorial and South Atlantic Ocean, in particular on the equatorial African margin and to a lesser degree in the Equatorial East Pacific Ocean. As discussed later (Sect. 3.1.2), these positive density residuals could be related to ecological factors such as seasonality.

It is possible to take into account seasonality based on an estimation of foraminiferal seasonal abundance (Malevich et al. 2019), or using the FAME module. This module predicts the mean $\delta^{18}Oc$ of a foraminifera sample constituted of a number of individuals by weighting in space (depth in the water column) and time (months) the oceanic conditions by the growth rate of each individual.

We decided to not directly develop seasonal calibration models for several reasons. First, we
want to predict annual surface density to be able to compare and evaluate numerical
climate models against annual surface density. Second, including seasonal signals in
foraminifera in our Bayesian models using sediment trap data (Malevich et al. 2019) or
seasonality and habitat depth using FAME (that uses the temperature dependence of growth
derived from culture experiments (Lombard et al., 2009)) would be a simplification that does
not consider factors such as light and nutrient availability. Third, even if it could potentially
improve the models for the present day calibration, although a hierarchical seasonal model
does not necessary show an increase in validation performance compared to the hierarchical
annual model (Malevich et al., 2019), this approach assumes that seasonality or habitat
depth would not change during past periods. Results using FAME demonstrate that
seasonality or habitat depth change during past periods (Roche et al., 2018). Therefore,
changes in seasonality and habitat depth could introduce additional uncertainties when
using a seasonal calibration model to predict past seasonal surface density. One possibility
would be to use simulation results for past periods to force the FAME module and create
past Bayesian calibration models between $\delta^{18}Oc$ and surface density that would take into
account ecological changes. However this would not be independent of climate models and
would lead to circular reasoning if the purpose is to use reconstructed density for
comparison and evaluation of past climate simulations.
We therefore adopt a different strategy. We use past isotope enable climate model
simulations for the pre-industrial (PI) and LGM periods to force the FAME module in order to
test within the "model world" if a PI Bayesian calibration (hierarchical design) between the
$\delta^{18}Oc$ and annual surface ocean density is stable with time and if the changes in foraminifera
growth season and habitat depth lead to additional uncertainties when applying a PI relation
to past annual predictions (LGM).

3.1.2 Testing the stability of the $\delta^{18}Oc$-density relation during past periods
Because the proposed approach to reconstruct ocean surface density uses the temperature
and $\delta^{18}Osw$ influence on the $\delta^{18}Oc$ signal, we investigated the potential evolution of the
$\delta^{18}Oc$-density relationship with time before applying this approach to past density
reconstructions. In particular, we investigated two questions: does the present day $\delta^{18}Osw$-
salinity relationship and its known past temporal evolution (Rohling, 2000, LeGrande and
Schmidt, 2011, Caley and Roche, 2015) significantly affect the density-$\delta^{18}Oc$ relation
evolution? Do ecological changes (foraminifera growth season and habitat depth)
significantly affect the density-$\delta^{18}Oc$ relation evolution?
We use numerical climate simulations (LGM and PI) of two isotope enabled numerical
climate models, iLOVECLIM and ECHAM5/MPI-OM, to address these questions. We calculate
the $\delta^{18}Oc$ signal based on the simulated $\delta^{18}Osw$ and ocean temperature for both PI and LGM
using the quadratic approximation of Kim and O'Neil (1997) given in Bemis et al. (1998). We
use the FAME module to predict the $\delta^{18}Oc$ values and account for foraminifera specific living
habitats in the water column and along the year as described in Roche et al. (2018). A
comparison of the simulated and observed core-top data $\delta^{18}Oc$ (Fig. 4) shows high
correlation ($R^2$ of 0.93 and 0.89 for ECHAM5/MPI-OM and iLOVECLIM respectively). The
slightly higher correlation with ECHAM5/MPI-OM and associated lower root mean square
error (RMSE) (Fig. 4) could be related to differences in climate models but also to the fact
that in the chosen configuration iLOVECLIM generated only annual $\delta^{18}Osw$ and ocean
temperature hydrographic data contrary to ECHAM5/MPI-OM that produces monthly
results. Therefore, the seasonality effect is only simulated by combining FAME and
ECHAM5/MPI-OM whereas the habitat depth effect is simulated in both experiments.
We tested this hypothesis by using yearly ECHAM5/MPI-OM values to compute the $\delta^{18}Oc$
and compared the results with those obtained with seasonal values (shown in Figure 4a) and
better assess the effect of seasonality. Results indicate a slight decrease of the $R^2$ of 0.02 and
a slight increase in RMSE of 0.06 when seasonality is not taken into account. These
differences are significant according to paired t-tests. Therefore, seasonality partly explains
the small difference between the results using ECHAM5/MPI-OM and iLOVECLIM. Lower
resolution of iLOVECLIM or other missing/biased processes in this model could also
contribute to this small difference.
Although climate models are not perfect, the observed high correlations demonstrate that 1)
these numerical climate models can be used to address our questions regarding the stability
of the $\delta^{18}Oc$-density relation during the past and 2) our hypothesis that planktonic
foraminifera $\delta^{18}Oc$ is mainly related to the surface ocean density, growth season and habitat
depth, with weak additional influence by biological processes (Sect. 1.) is valid.

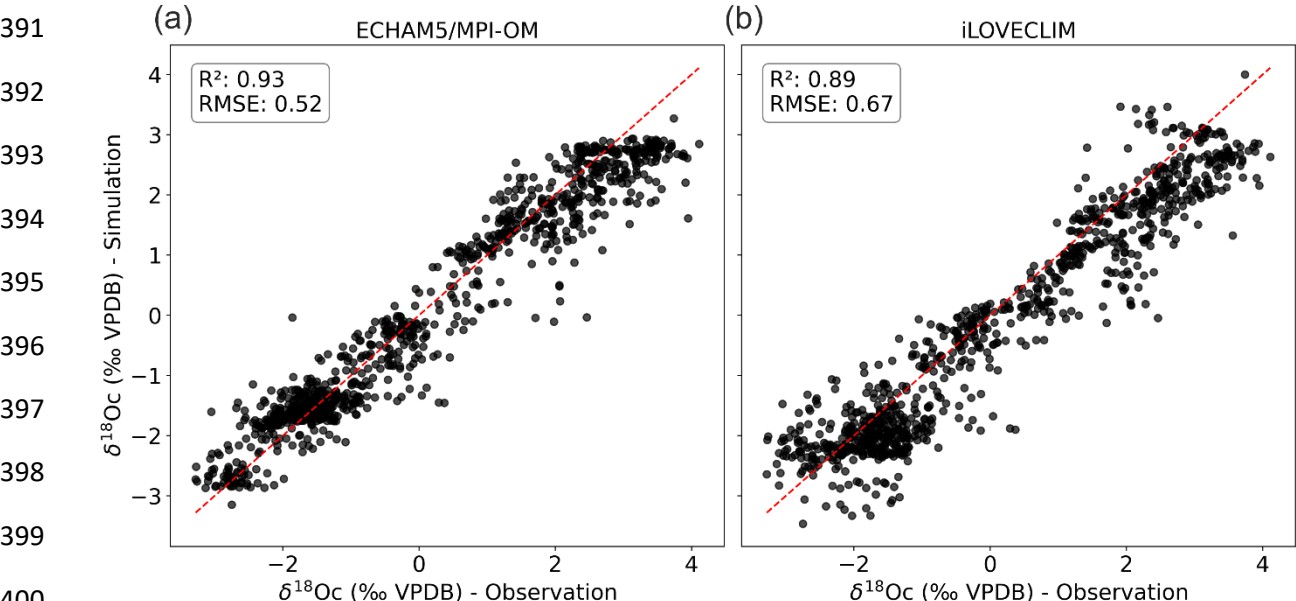


Figure 4: comparison between simulated PI foraminifera $\delta^{18}Oc$ (FAME module forced with
(a) ECHAM5/MPI-OM and (b) iLOVECLIM climate model hydrographic data) and observed LH
core-top $\delta^{18}Oc$ data. The 1:1 line is indicated.
We developed two PI Bayesian calibrations (hierarchical design) between the $\delta^{18}Oc$ and
annual surface ocean density based on FAME forced by ECHAM5/MPI-OM and iLOVECLIM
hydrographic data (Fig. 5a). These Bayesian calibration models are comparable to the
poly1_hier Bayesian calibration model of Fig. 1. We then used the LGM simulations to force
FAME and produce $\delta^{18}$Oc LGM values comparable to those that could be measured in a
marine sediment core (but in the model world). We can use these $\delta^{18}$Oc LGM values and the
PI Bayesian calibrations to predict the ocean surface density at the LGM. We can then
compare the density reconstructed from the $\delta^{18}$Oc values to the density simulated directly at
the LGM by ECHAM5/MPI-OM and iLOVECLIM. This furnish a test in the model world
regarding the stability of the $\delta^{18}$Oc-density relation during the past.
Interestingly, the observed (Fig. 1) and simulated (Fig. 5b and e) density residuals (predicted
- observed) are overall in good agreement for both PI ECHAM5/MPI-OM and iLOVECLIM
experiments in terms of qualitative changes (positive or negative residuals) (Fig. 5b and e
and Fig. 1). Nonetheless, differences for some regions in terms of magnitude of the residual
values exist between ECHAM5/MPI-OM and iLOVECLIM experiments. We observe high
positive residuals in the Equatorial and South Atlantic Ocean in the ECHAM5/MPI-OM
experiment, in particular on the equatorial African margin and in the Equatorial East Pacific
Ocean. As discussed before (Sect. 3.1.1), these positive density residuals are also visible in
the observations (Fig. 1f). We attribute these high positive residuals in ECHAM5/MPI-OM
(Fig. 5b) that better fit the observations (Fig. 1f) to a seasonality effect because seasonality is
only taken into account in ECHAM5/MPI-OM experiment. Negative residuals previously
discussed in upwelling regions are visible in simulated residuals but with lower magnitude in
comparison to observations (Fig. 1f and 5b and e). This could be related to the fact that
upwellings are not well simulated in the two experiments or to the role of secondary
environmental parameters such as nutrients and light penetration.
We apply the PI annual Bayesian calibration to the simulated LGM $\delta^{18}$Oc after a correction of
1.0‰ of LGM $\delta^{18}$Osw values (value added at LGM for the ECHAM5/MPI-OM and iLOVECLIM
experiments, Caley et al., 2014, Werner et al., 2016) to account to a change of the global
oceanic $\delta^{18}$Osw signal due to the increased LGM ice sheets. This yields a prediction of the
LGM surface ocean density that we can compare to the directly simulated LGM surface
density in both experiments. We calculate the density residual at the LGM (density
reconstructed from the $\delta^{18}$Oc values - density simulated directly at the LGM). Finally, we
calculate the density residuals anomaly between LGM and PI as:  density residuals at LGM -
density residuals at PI (Fig. 5c and f). This allows us to investigate the additional uncertainties
linked to the evolution of the density-$\delta^{18}$Oc relation (Fig. 5c and f).

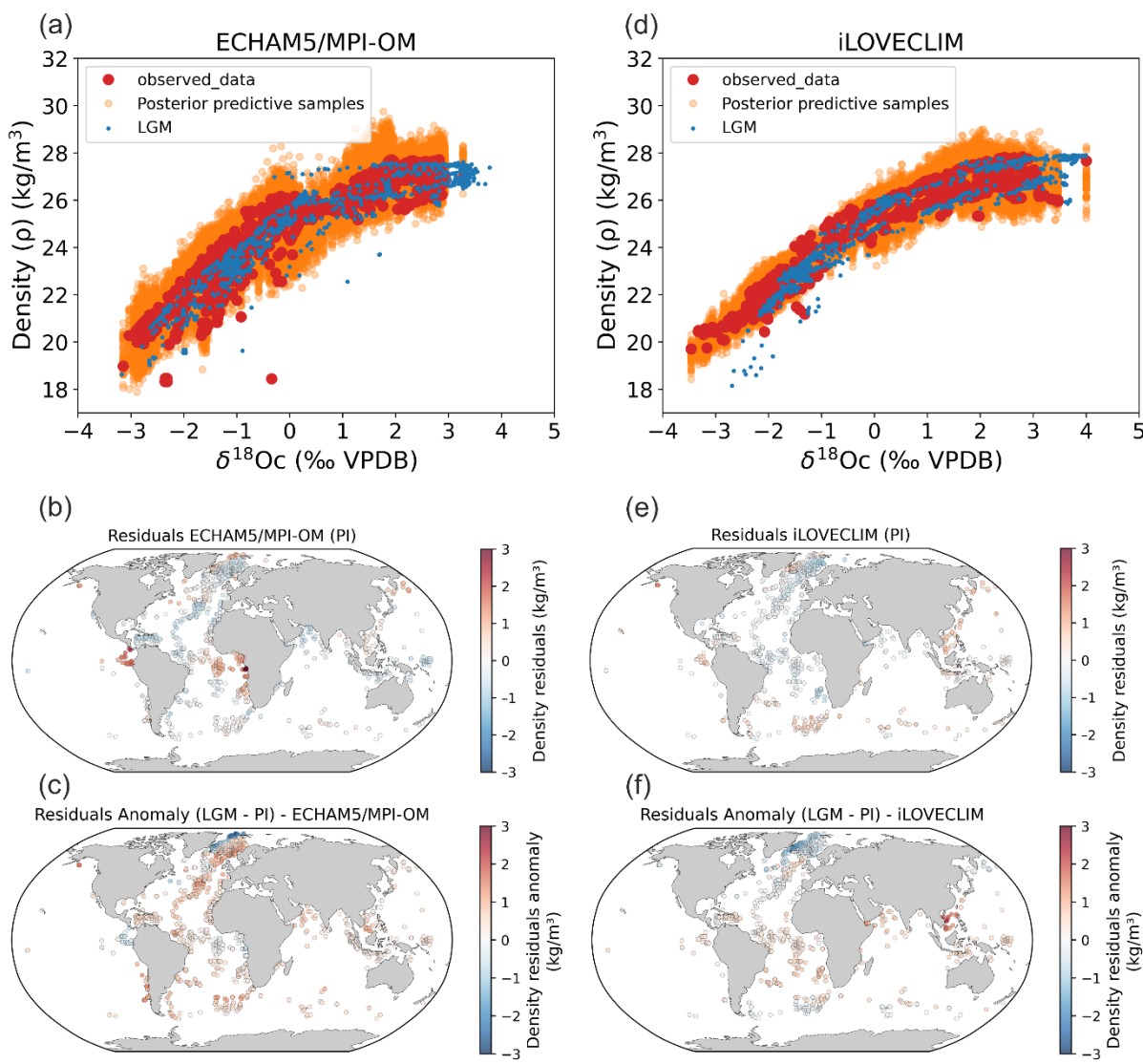


Figure 5: Stability of foraminifera δ¹⁸Oc-density relations between PI and the LGM calculated
with FAME and forced by global ECHAM5/MPI-OM (left panels, Werner et al., 2016) and
iLOVECLIM (right panels, Caley et al., 2014) hydrographic data. (a) and (d) PI Bayesian
regression models between foraminifera δ¹⁸Oc and annual surface density. Data in the PI
experiments have been selected at the same locations as observations (Fig. 1). Posterior
predictive samples and the LGM δ¹⁸Oc-density relation (LGM) are visible. (b) and (e) Density
residuals (predicted - observed) for the PI experiments. (c) and (f) Density residuals anomaly
between LGM and PI. Results for the Mediterranean Sea have been excluded because of its
difficulty to be simulated and inconsistency between the two model simulations because of
their different grid resolutions. Annual mean temperature and δ¹⁸Osw were used for the
iLOVECLIM experiment whereas monthly temperature and δ¹⁸Osw were used for the
ECHAM5/MPI-OM experiment.

The surface density residuals anomalies (LGM - PI) are overall rather close to 0 except in the
Nordic Seas region (north of 40°N in the Atlantic). For the following analyses we do not
consider the North Indian Ocean for iLOVECLIM. Indeed, this region is affected by a well-
known bias of this climate model due to a shift of the African precipitation regions from the

west to the east of the continent, leading to much less saline waters than presently observed
(and unrealistically depleted $\delta^{18}Osw$) in the North Indian Ocean (Roche and Caley, 2013).
Higher residuals anomaly in Nordic Seas region could be associated with difficulty in
simulating the $\delta^{18}Osw$-salinity relation evolution related to ice-sheets and sea ice changes
and/or to foraminifera ecological changes between LGM and PI. We also observe in this
region larger surface density residuals anomalies (LGM – PI) with ECHAM5/MPI-OM than
with iLOVECLIM (Figure 5c and f). This can be explained by different simulated sea ice
coverage in ECHAM5/MPI-OM compared to iLOVECLIM. Indeed, the Nordic Seas is the
region with the largest difference between the two model simulations of modeled annual SST
below 0°C (https://doi.org/10.5194/egusphere-2025-2459-AC2). Temperature is used to
calculate the $\delta^{18}Oc$ signal, ocean density and to force the FAME module. Any temperature
difference in the Nordic Seas thus affects density reconstructions and hence the density
residuals (Figure 5c and f).

To further investigate in a more quantitative way if the use of the PI bayesian calibration to
predict LGM surface density introduces additional uncertainties, we compare probability
distributions of surface density residuals anomaly (LGM - PI) using violin and box plots to the
95% confidence interval (CI) of the PI bayesian calibration (Fig. 6). We consider each
foraminifera species separately. Global results indicate for the *G. ruber* and *T. sacculifer*
species that 1) the 5th to 95th percentile and interquartile range of the surface density
residuals anomaly is well inside the 95% CI of the PI bayesian calibration for both
ECHAM5/MPI-OM and iLOVECLIM experiment and 2) high probability and median values are
close to 0 (Fig. 6a and c). This is not the case for *G. bulloides*, *N. incompta*, and for *N.*
*pachyderma*.
When the Nordic Seas region is removed, results indicate that for all the foraminifera
species, the interquartile range of the surface density residuals anomaly is well inside the
95% CI of the PI bayesian calibration for both experiments (ECHAM5/MPI-OM and
iLOVECLIM). High probability and median values are closest to 0 (Fig. 6b and d). The 95% CI
of the PI bayesian calibration is closest to the 5th to 95th percentile range of the surface
density residuals anomaly.

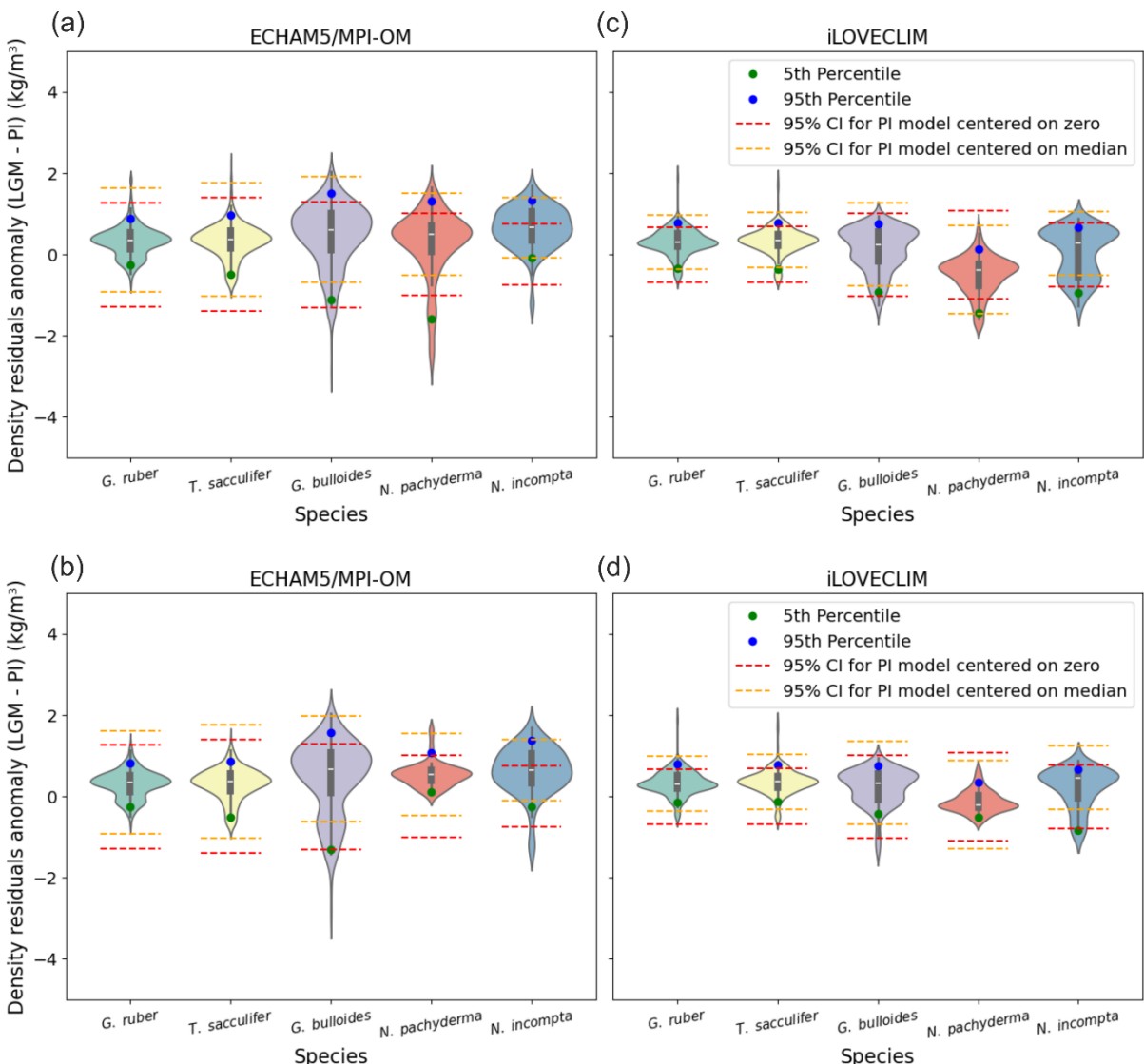

Figure 6: probability distributions of surface density residuals anomaly (LGM - PI) for
ECHAM5/MPI-OM and iLOVECLIM, for global data (a and c), and without the Nordic Seas and
northern North Atlantic (north of 40°N) (b and d). North Indian Ocean data for iLOVECLIM
have been removed in both cases.

We conclude based on our tests that the use of a Bayesian calibration model (hierarchical
design) to predict annual surface density during past periods (with the example here of the
LGM climate) is valid globally within the explicitly estimated uncertainty in calibration model
parameters, except for the Nordic Seas region.

3.2 Reconstruction of past ocean surface absolute density

To reconstruct past ocean surface absolute density based on foraminifera $\delta^{18}O_c$ values that
have been corrected from the $\delta^{18}O_{sw}$ ice effect, an additional correction is necessary.
Indeed, it is necessary to account for mean ocean density changes related to ocean volume
changes that affect mean ocean salinity. Without this additional correction, the ocean
density reconstructed corresponds to density changes linked to hydrographic changes in SST
and SSS.

To determine the mean ocean density change related to the change in ocean volume at LGM
we used model simulation results (ECHAM/MPI-OM and iLOVECLIM) and added or removed
1 psu salinity (Duplessy et al., 1991) in global salinity outputs. Note that adding 1 psu of
salinity at LGM in climate model simulations has only small effects on ocean dynamics.
Indeed, the effect is due to the small non-linearity in the sea-ice freezing, hence generating
small differences in regions of sea ice and deep water formation. We have tested it in new
simulations performed with the iLOVECLIM model and found the dynamical effect of a 1 psu
salinity change in the regions we are analyzing to be very small (not shown).
Both model simulations agree and yield a mean ocean salinity effect on density of 0.776 ($\sigma$ =
0.02) for ECHAM/MPI-OM and 0.772 ($\sigma$ = 0.02) for iLOVECLIM.  We also performed a
calculation to estimate this effect based on observations (reference state based on present
day observations and LGM state based on Tierney et al., 2020 for SST and Duplessy et al.,
1991 for SSS) and found very consistent results (https://doi.org/10.5194/egusphere-2025-
2459-AC1).
Therefore, the additional correction that is necessary to reconstruct past ocean surface
absolute density at the LGM is estimated to be equal to + 0.77.
3.3 LGM annual surface density reconstruction
We applied the poly1_hier calibration model to the LGM and LH $\delta^{18}$Oc foraminifera
database, excluding the Nordic Seas region, after subtraction of 1.0 ‰ from LGM $\delta^{18}$Oc
values (Labeyrie et al. 1987; Schrag et al., 1996; Schrag et al., 2002; Adkins et al., 2002;
Duplessy et al., 2002) in order to reconstruct LGM and LH annual surface density. Absolute
LGM annual surface density was calculated by adding 0.77 to density changes linked to
hydrographic changes in SST and SSS. The benefit of our Bayesian model is the possibility to
propagate uncertainty from calibration into predictions of past climate conditions (Fig. 7).













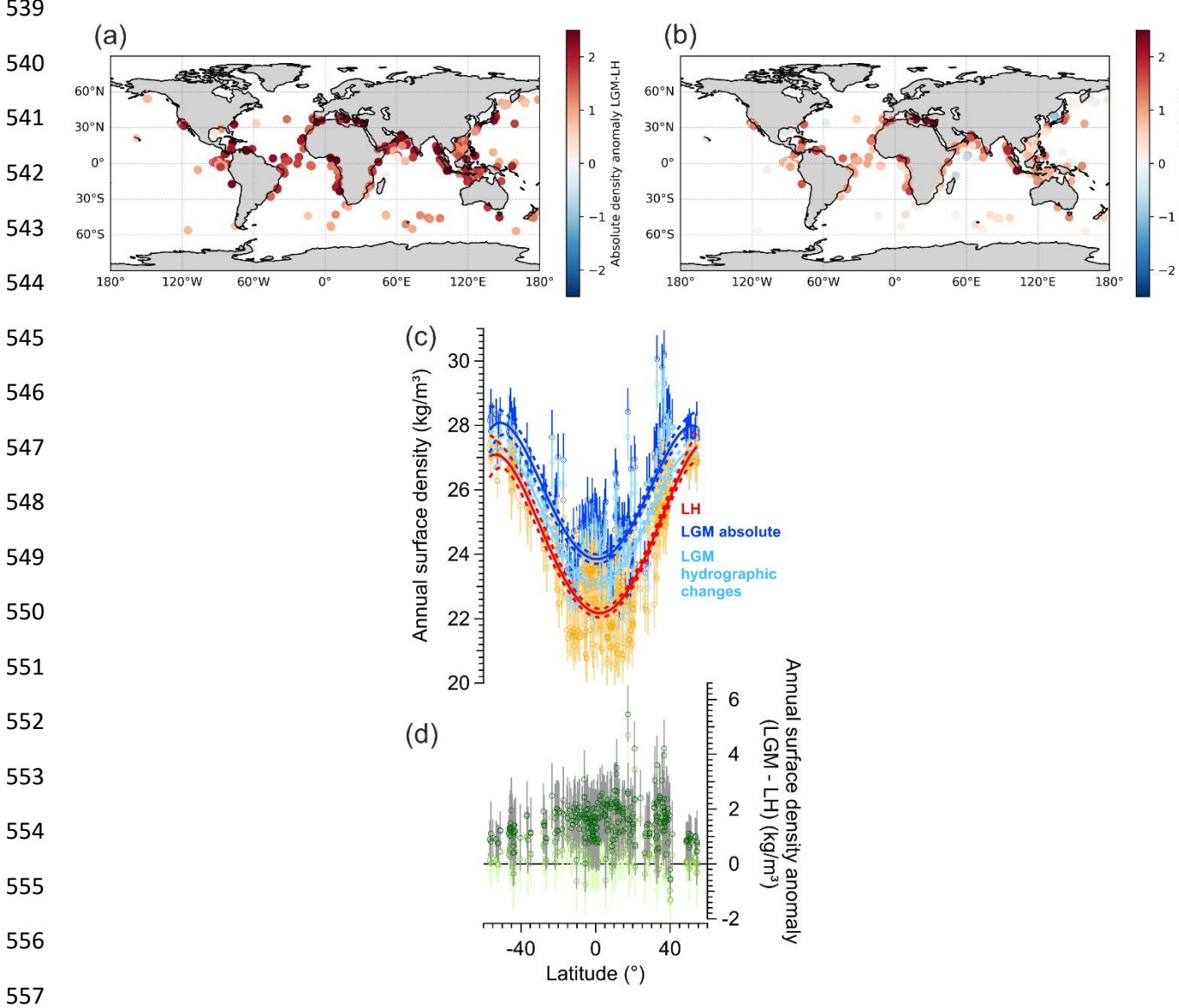

Figure 7: reconstructions of LGM and LH annual surface ocean density from foraminifera
$\delta^{18}Oc$. (a) Spatial distribution of the LGM - LH absolute density anomaly. (b) Spatial
distribution of the LGM - LH density changes due to hydrographic changes in SST and SSS. (c)
Meridional gradient of reconstructed surface annual LGM density (absolute density in dark
blue, density due to hydrographic changes in light blue) and comparison with LH
reconstructions (red and orange colors). Error bars for each data point represent the 68 %
C.I. A polynomial fit (5th degree) and associated 95% confidence bands are shown as solid
resp. dashed lines. (d) Meridional gradient of reconstructed density anomaly (LGM - LH) for
absolute density in dark green and density due to hydrographic changes in light green and
associated 68 % C.I (grey lines).

Ocean surface density increases globally during the LGM in agreement with colder SST
(MARGO, 2009; Tierney et al., 2020b) and increases global salinity (Duplessy et al., 1991;
Adkins et al., 2002) (Fig. 7a). We also observe stronger LGM density value changes at low
latitudes compared to mid latitudes (Fig. 7b, c and d). This is probably the result of the LGM
cooling (MARGO, 2009; Tierney et al., 2020b) in combination with a reduction of the
intensity of low latitudes hydrological cycle (Kageyama et al., 2021), whereas higher latitudes
are already close to ocean density maximum. Further regional analyses of ocean surface
density and comparison with numerical climate models are presented in Barathieu et al. in
prep.

4. Conclusions
We developed three Bayesian regressions to model the relationship between the calcite
oxygen isotopic composition of planktonic foraminifera, $\delta^{18}Oc$, and annual mean surface
density, $\rho$. This allowed us to explicitly estimate the uncertainty in calibration model
parameters. We find that predictive performance of the model improves when we account
for inter-species specific differences. Before applying this model to past density
reconstructions, we used results of isotope enabled climate model simulations for PI and
LGM time periods to force the FAME module. We then investigated the additional
uncertainties that could be introduced by potential evolution of the $\delta^{18}Oc$-density
relationship with time. It could be caused by changes in the $\delta^{18}Osw$-salinity relationship or
by foraminifera ecology. We demonstrate that additional uncertainties are weak and that
our approach is valid (except for the Nordic Seas region), within propagated uncertainty
from calibration into predictions of past climate conditions.
By applying our Bayesian regression hierarchical model to LGM and LH $\delta^{18}Oc$ foraminifera
databases, we reconstructed LGM and LH annual surface density and found stronger LGM
density value changes at low latitudes compared to mid latitudes. The logical next step will
be to compare globally and in more detail (regional scale) our quantitative annual surface
density reconstruction with densities obtained by numerical climate model simulations
during the LGM. This will be used to evaluate these climate models in their ability to
simulate this parameter during this extreme climatic period (Barathieu et al., in prep). The
quantification of density together with the estimation of uncertainties could also be used for
data assimilation approaches, allowing local paleoclimate proxy information to be used to
infer global climate metrics (Tierney et al., 2020a).
We demonstrate that our approach is valid to quantitatively reconstruct annual surface
density during one of the coldest climates of the Quaternary period. We also demonstrate
this for the mid Holocene and last interglacial periods (Appendix B). Hence, our calibration
has great potential to be applied to other past periods and to reconstruct past temporal
evolution of ocean surface density downcore during the Quaternary. Under very extreme
climates outside the Quaternary (Appendix B) and in ocean basins characterized by anti-
estuary circulation, like the current Mediterranean Sea and Red Sea, our calibration could
provide density estimates with larger uncertainty, a point that requires further
investigations.
Finally, our calibration method to quantitatively reconstruct past ocean surface density is
stable with time. A combination with existing calibration methods to reconstruct past SST
could lead to a "time stable" method to quantitatively reconstruct past SSS over the
Quaternary, contrary to the use of the $\delta^{18}O_{sw}$-SSS approach. Before realized SSS
reconstructions, further investigations and calculation of uncertainties are necessary for this
potential new method. This is clearly a way forward as SSS is a crucial parameter that can
provide insights into hydrological cycle dynamics and its evolution.
























# Appendices

## Appendix A. Detailed Prior Specifications

Below we provide the exact prior definitions and hyperparameter settings for each of the three Bayesian models. In the following, $\rho$ denotes annual mean surface density, and $\delta c$ represents $\delta^{18}Oc$. Let $E[\rho]$ and $var(\rho)$ be the sample mean and variance of $\rho$, respectively, and let $var(\delta c)$ be the sample variance of $\delta c$.

1.  **First-Degree Polynomial (Pooled)**

$$\rho \sim N(\mu, \sigma^2)$$
$$\mu = \beta_0 + \beta_1 \delta_c.$$

We chose weakly informative and data-adaptive priors, meaning they center around observed mean/variance but are broad enough to allow for uncertainty.

$$\beta_0 \sim N(E[\rho], 2.5\sqrt{var(\rho)}), \quad \beta_1 \sim N\left(0, 2.5\sqrt{\frac{var(\rho)}{var(\delta_c)}}\right), \quad \sigma \sim Exp(\sqrt{var(\rho)^{-1}}).$$

2.  **Second-Degree Polynomial (Pooled)**

$$\rho \sim N(\mu, \sigma^2)$$
$$\mu = \beta_0 + \beta_1 \delta_c + \beta_2 \delta_c{}^2.$$

We set the priors to

$$\beta_i \sim N(0, 6.08^2) \text{ for } i \in \{0, 1, 2\}, \quad \sigma \sim Exp(\sqrt{var(\rho)^{-1}}).$$

Here, the normal priors were chosen to ensure that 90 % of the prior mass for each $\beta_i$ lies within [-10, 10].

3.  **First-Degree Polynomial (Hierarchical)**

$$\rho \sim N(\mu_s, \sigma_s{}^2)$$
$$\mu_s = \beta_{s,0} + \beta_{s,1} \delta_c$$

where each species $s$ has its own slope and intercept. These species-level parameters share hyperpriors:

**Species-Level Parameters**

$$\beta_{s,i} \sim N(\nu_i, \kappa_i{}^2), \quad i \in \{0, 1\}, \quad \sigma_s \sim Exp(\lambda_s).$$

**Hyperpriors**


$$\nu_0 \sim N(E[\rho], 10), \qquad \nu_1 \sim N(0, 10)$$

$$\kappa_0 \sim Exp(2.5\sqrt{var(\rho)}), \qquad \kappa_1 \sim Exp(2.5\sqrt{\frac{var(\rho)}{var(\delta_c)}}),$$

$$\lambda_s^{-2} \sim LogNormal(0, 1).$$


683

688

## Appendix B. Application of our calibration to other past periods

Our study is focused on the LGM but it is interesting to examine if our results remain valid
for other climate periods. In this appendix, we present tests using isotope-enabled model
runs representing different past climate conditions in order to demonstrate that additional
uncertainties due to the evolution of the $\delta^{18}O_c$-density relationship with time are globally
small and that the new calibration has great potential to reconstruct the past temporal
evolution of ocean surface density over the Quaternary period.

In addition to the LGM time period investigated in our study, we tested the Mid Holocene
(MH) period and the last interglacial period (LIG) (Figs. B1 and B2). Results clearly indicate
a strong stability of foraminifera $\delta^{18}O_c$-density relations between MH, LIG and the PI, that is
a very weak influence of the changes in the $\delta^{18}O$/Salinity relation or foraminifer ecology (i.e.
habitat depth and growing season) on final density predictions (Figs. B1 and B2).















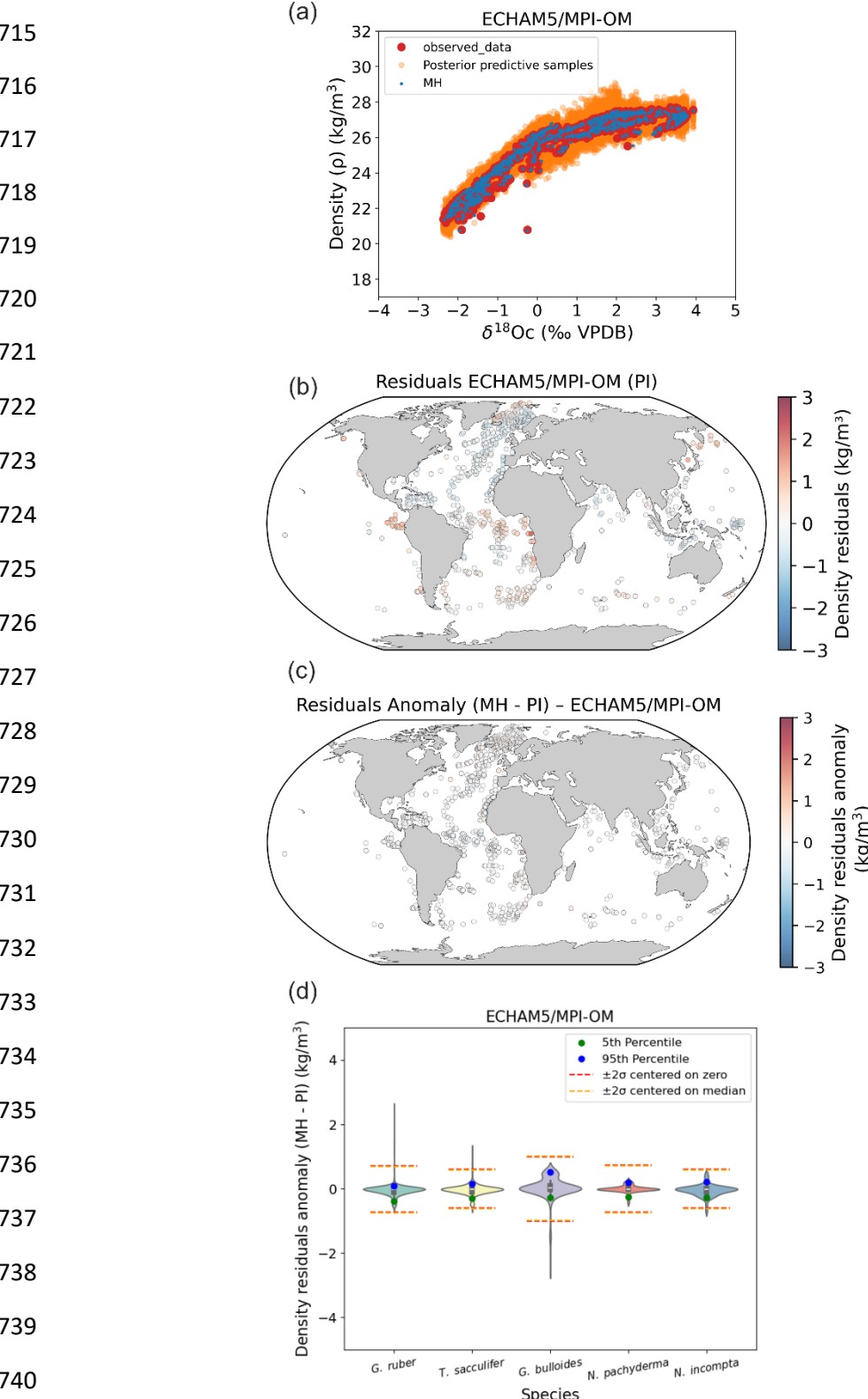

Fig. B1: Stability of foraminifera $\delta^{18}Oc$-density relations between PI and the MH calculated with FAME and forced by global AWI-ESM-2.1-wiso (Shi et al., 2023) hydrographic data. (a) PI Bayesian regression models between foraminifera $\delta^{18}Oc$ and annual surface density. Posterior predictive samples and the MH $\delta^{18}Oc$-density relation (MH) are visible. (b) Density residuals (predicted - observed) for the PI experiments. (c) Density residuals anomaly between MH and PI. (d) Probability distributions of surface density residuals anomaly (MH - PI) without Nordic Seas and northern North Atlantic (north of 40°N).

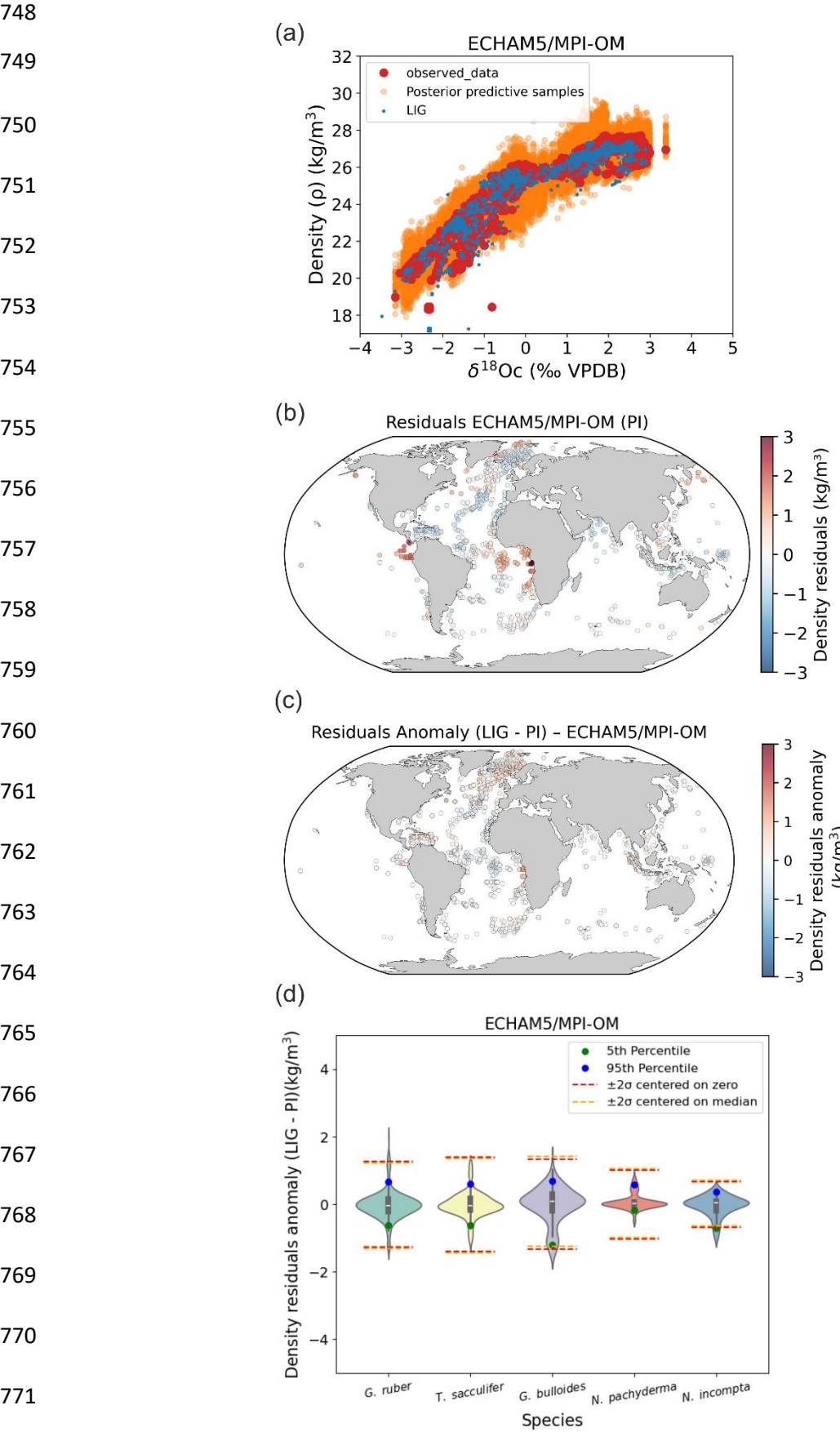

Fig. B2: Stability of foraminifera $\delta^{18}$Oc-density relations between PI and the LIG calculated
with FAME and forced by global ECHAM5/MPI-OM (Gierz et al., 2017) hydrographic data.
(a) PI Bayesian regression models between foraminifera $\delta^{18}$Oc and annual surface density.
Posterior predictive samples and LIG $\delta^{18}$Oc-density relations (LIG) are visible. (b) Density
residuals (predicted - observed) for the PI experiments. (c) Density residuals anomaly
between LIG and PI. (d) Probability distributions of surface density residuals anomaly (LIG -
PI) without Nordic Seas and northern North Atlantic (north of 40°N).

Applying our calibration to past climates (and taking into account foraminifer ecological
changes) provides density predictions that remain within the uncertainties of the calibration,
as demonstrated for the LGM, MH and LIG time periods. These time periods correspond to
extreme climate configurations over the Quaternary period as shown on Fig. B3, so the new
calibration can be reliably applied to reconstruct the past temporal evolution of ocean surface
density over the entire Quaternary (last 2.6 Ma).

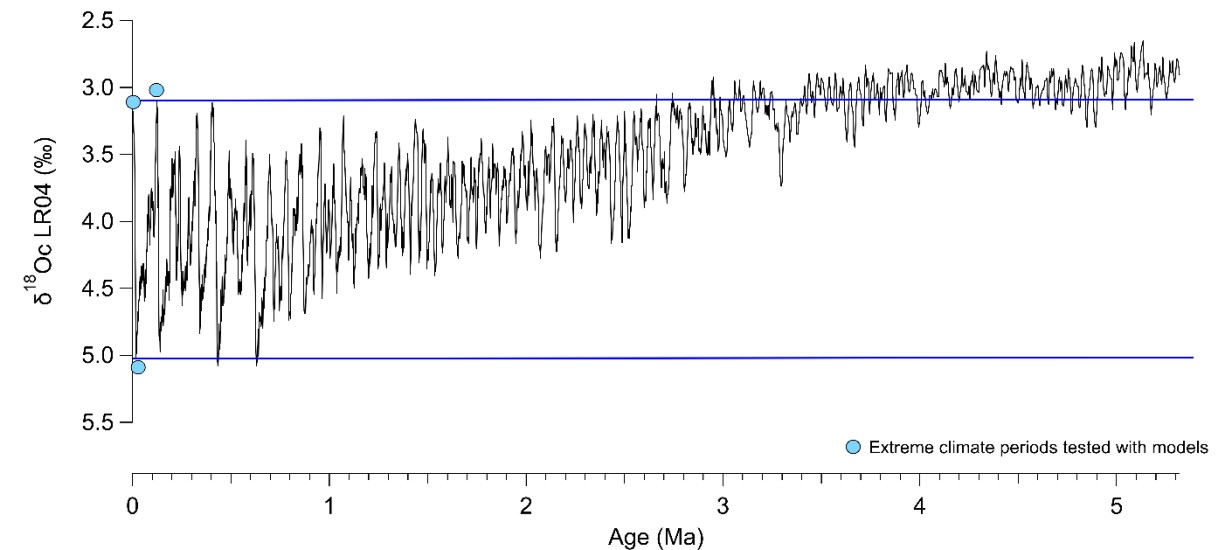

Fig. B3: $\delta^{18}$O benthic foraminifera curve (LR04, Lisiecki and Raymo, 2005) as a proxy of ice
volume and deep ocean temperature changes, used here to select extreme climatic periods
(colder and more arid glacial periods versus warmer and more humid interglacial periods).
Extreme climate periods tested with isotope-enabled model runs representing the mid-
Holocene, LIG and LGM are represented by blue dots. Blue lines indicate the range of
extreme climate conditions investigated with our climate simulations tests.
Nonetheless, under very extreme climates outside the Quaternary period (Fig. B3) and in
ocean basins characterized by anti-estuary circulation, like the current Mediterranean Sea and
Red Sea, our calibration could provide density estimates with larger uncertainty, a point that
requires further investigations.


**Code and data availability**

The Python code for Bayesian calibration models is freely available at the following repository: https://github.com/nicrie/density_uncertainty**.** Core top data used for this analysis are from Malevich et al. 2019 and are available at https://agupubs.onlinelibrary.wiley.com/doi/10.1029/2019PA003576. LGM and LH $\delta^{18}$Oc dataset are available at doi:10.5194/cp-10-1939-2014-supplement for Caley et al., 2014, at https://doi.org/10.1594/PANGAEA.894229 for Waelbroeck et al., 2014 and at https://doi.org/10.1594/PANGAEA.920596 for Tierney et al., 2020b. The additional LGM and LH $\delta^{18}$Oc dataset is available at the following repository: https://github.com/nicrie/density_uncertainty.

**Author contributions**

TC and DR initially designed the study. TC developed the study. NR and TC developed the Bayesian calibration models. MW provided ECHAM5/MPI-OM model outputs. CW furnished the new $\delta^{18}$Oc dataset. TC analysed the results with contribution and discussion of all co-authors. TC produced the figures and wrote the article with input from all co-authors.

**Competing interests**

The contact author has declared that none of the authors has any competing interests.

**Acknowledgements**

T.C. is supported by CNRS Terre & Univers. This study has been conducted using E.U. Copernicus Marine Service Information; https://doi.org/10.48670/moi-00051.

**Financial support**

This research was supported by the ANR HYDRATE project (grant no. ANR-21-CE01-0001) of the French Agence Nationale de la Recherche.

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
