# Peer review of "Past Ocean surface density from planktonic foraminifera calcite $\delta^{18}\text{O}$"

_EGUsphere, 2025_

## Author Response (AR1)

**Response to reviewers**

We are very grateful for the constructive feedbacks provided by the 3 reviewers that helped us to significantly improve our manuscript. Below, we present a point-by-point response to all the individual comments raised.

We have carefully considered and addressed all comments and suggestions of the reviewers. This led to additional text and revised Figures in the revised version according to reviewer suggestions. We also added a new Appendix B: "Application of our calibration to other past periods" in the revised version, containing the additional tests we conducted during the review process for the MH and LIG time periods which allow us to contextualize our results for different climate periods. The Code and Data Availability section has been updated to indicate that the Python code for the Bayesian calibration models and the new dataset are freely available at the following repository: https://github.com/nicrie/density_uncertainty.

In our response to the reviewers' comments, we present some results of data-model comparison from the paper of Barathieu et al., that will be submitted very soon, to show that the models used in our study (MPI and iLOVECLIM) exhibit significant (p-value <0.05) and strong correlation ($R^2$>0.5) with our reconstructed densities for the PI and LGM.

Reviewer comments are shown in red color, with our responses in black.

Changes in the manuscript version (marked-up manuscript version) are in red color.

Sincerely,

Thibaut Caley on behalf of all co-authors

**Response to Reviewer 1**

Caley et al. calibrate planktic $\delta^{18}O$ from core tops to surface density and assess the uncertainty via Bayesian modelling. The authors test the calibration with the results of isotope enabled models and apply the method to foraminiferal $\delta^{18}O$ values from the Last Glacial Maximum and the late Holocene. The attempt to translate $\delta^{18}O$ directly into density is certainly worthwhile since we have probably an order of magnitude more $\delta^{18}O$ data available compared to combined $\delta^{18}O$/temperature reconstructions. However, the paper has methodological and transparency issues that need to be addressed.

**1. Representation of mean ocean density.** There are surface density changes related to local/regional SST and SSS changes and mean ocean density changes related to ocean volume. Part of the local/regional density changes will be related to mean ocean salinity due to volume changes with sea level. For example, Duplessy et al. (1991) estimated that the smaller LGM ocean volume led to ~1 psu higher salinity (and hence a significantly higher global ocean LGM density). In principle, foraminiferal $\delta^{18}O$ contains information on sea level via the ice effect (albeit with a higher slope as usual evaporation -precipitation changes). The $\delta^{18}O$ ice effect, however, is removed before the LGM density reconstruction. Can the LGM reconstructions really reflect absolute density or have the authors rather reconstructed the density changes due to local/regional changes in SST and SSS? Perhaps I am missing something here, but in my view the mean ocean density changes corresponding to mean ocean salinity changes due to ice/ocean volume changes have to be added to the LGM values since the used foraminiferal isotope data do not contain this information and the method does not account for it. This point requires further clarification.

We thank the reviewer for this thoughtful comment and we agree with the reviewer that this point requires further clarification in our paper. What we reconstructed are indeed the density changes due to the hydrographic changes in SST and SSS (the local/regional SST and SSS changes mentioned by the reviewer). To determine mean ocean density changes related to ocean volume we used model simulations results (ECHAM/MPI-OM and iLOVECLIM) and added or removed 1 psu salinity in global salinity outputs. Note that adding or removing 1 psu of salinity at LGM in climate model simulations have only small effects on ocean dynamic. Indeed, the effect is due to the small non-linearity in the sea-ice freezing, hence generating small differences in regions of sea ice and deep water formation. We have tested it in new simulations performed with the iLOVECLIM model and found the dynamical effect of that 1 psu in the regions we are analyzing to be very small (not shown). We also added or removed 1 ‰ in the ocean $\delta^{18}Osw$ reflecting the global mean ice-sheets contribution.

[Figure]

Figure a1: $\delta^{18}$Oc-density relationship based on observations (red), PI simulations (grey) and LGM simulations without (blue) and with a + 1 psu salinity increase (cyan).

[Figure]

Figure a2: Same as Figure a1, but the LGM simulations additionally include a +1 ‰ $\delta^{18}$Osw offset (cyan).

In Figure a1 we observe how a change in + 1 (cyan color) or − 1 (blue color) psu salinity in model simulations affect the surface density at the LGM and so the $\delta^{18}$Oc-density relation. In Figure a2, the cyan colour data show the $\delta^{18}$Oc-density relation at the LGM (with + 1 psu salinity and +1 ‰ $\delta^{18}$Osw) contrary to Figure a1 that only contain + 1 or − 1 psu salinity in model simulations. The δ¹⁸Oc-density relation at the LGM (with + 1 psu salinity and +1 ‰
δ¹⁸Osw, cyan colour) can be used to estimate LGM density. To reconstruct absolute density
at the LGM with the observed relation (red color on Figures a1 and a2), we can remove the
δ¹⁸O ice effect (1‰) but need to add mean ocean density changes corresponding to mean
ocean salinity changes due to ocean volume changes (Figure a1). If not added, we indeed
reconstruct the density changes due to hydrographic changes in SST and SSS (the
local/regional SST and SSS changes mentioned by the reviewer).

We calculated mean ocean density changes corresponding to mean ocean salinity changes
due to ocean volume changes by adding or removing +1 psu of salinity during the LGM in
model simulations (Figure b).

[Figure]

Figure b: surface ocean anomaly of density when +1 psu is added or removed during the
LGM at the location of our core top observations used in our calibration

Figure b shows the surface anomaly of density when +1 psu is added or removed during the
LGM at the location of our core top observations used in our calibration. Both model
simulations agree and yield a mean ocean salinity effect on density of 0.776 (σ = 0.02) for
ECHAM/MPI-OM and 0.772 (σ = 0.02) for iLOVECLIM.

We also performed a calculation to estimate this effect based on observation (reference
state based on present day observation and LGM state based on Tierney et al., 2020 for SST
and Duplessy et al., 1991 for SSS). To calculate uncertainties, we use a bootstrap simulation
approach (uncertainties of ± 0.1 g/kg for salinity and of ± 0.2 °C for temperature) to estimate
the isolated effect of increased salinity on surface seawater density under Last Glacial
Maximum (LGM) conditions (10,000 samples of temperature and salinity anomalies are
randomly drawn using a uniform distribution to reflect plausible variability).

Reference State Definition: Modern ocean conditions are used as a baseline, with surface
salinity of 34.7 g/kg and temperature of 13.85 °C.

Perturbation Parameters:  A salinity anomaly of +1 g/kg ± 0.1 is applied (Duplessy et al.,
1991). A surface temperature anomaly of −2.9 °C ± 0.2 simulates cooler LGM conditions
(Tierney et al., 2020).

Results of mean surface density anomaly corresponding to mean ocean salinity changes due
to ocean volume changes (LGM context): 0.772 kg/m³ 95% confidence interval: (0.698 to
0.845) kg/m³.

Results and uncertainties are in very good agreement with model simulations results (Figure
b).

**Therefore, depending what we want to reconstruct, absolute density or density changes**
**linked to hydrographic changes in SST and SSS (the local/regional SST and SSS changes**
**mentioned by the reviewer), we need or not need to have an additional correction. For**
**absolute density, an additional correction is necessary, estimated to be equal to 0.77.**

We therefore added a new section "Reconstruction of past ocean surface absolute density"
to the text before section "3.2 LGM annual surface density reconstruction" in which we
explain that around +0.77 needs to be added to obtain absolute surface density values.

We added in lines 496-518: "3.2 Reconstruction of past ocean surface absolute density

To reconstruct past ocean surface absolute density based on foraminifera $\delta^{18}Oc$ values that
have been corrected from the $\delta^{18}Osw$ ice effect, an additional correction is necessary.
Indeed, it is necessary to account for mean ocean density changes related to ocean volume
changes that affect mean ocean salinity. Without this additional correction, the ocean
density reconstructed corresponds to density changes linked to hydrographic changes in SST
and SSS.

To determine the mean ocean density change related to the change in ocean volume at LGM
we used model simulation results (ECHAM/MPI-OM and iLOVECLIM) and added or removed
1 psu salinity (Duplessy et al., 1991) in global salinity outputs. Note that adding 1 psu of
salinity at LGM in climate model simulations has only small effects on ocean dynamics.
Indeed, the effect is due to the small non-linearity in the sea-ice freezing, hence generating
small differences in regions of sea ice and deep water formation. We have tested it in new
simulations performed with the iLOVECLIM model and found the dynamical effect of a 1 psu
salinity change in the regions we are analyzing to be very small (not shown).

Both model simulations agree and yield a mean ocean salinity effect on density of 0.776 (σ =
0.02) for ECHAM/MPI-OM and 0.772 (σ = 0.02) for iLOVECLIM.  We also performed a
calculation to estimate this effect based on observations (reference state based on present
day observations and LGM state based on Tierney et al., 2020 for SST and Duplessy et al.,
1991 for SSS) and found very consistent results (https://doi.org/10.5194/egusphere-2025-
2459-AC1).

Therefore, the additional correction that is necessary to reconstruct past ocean absolute
density at the LGM is estimated to be equal to + 0.77."

In part 3.2 we now propose the reconstruction of absolute density together with the density
changes due to hydrographic changes in SST and SSS (the local/regional SST and SSS changes
mentioned by the reviewer) for comparison and discussion (including a revised version of
Figure 7). Figures 5 and 6 have been also revised to take into account mean ocean density
changes related to ocean volume between iLOVECLIM and ECHAM/MPI-OM.

[Figure]

Revised Figure 5: Stability of foraminifera δ18Oc-density relations between PI and the LGM
calculated with FAME and forced by global ECHAM5/MPI-OM (left panels, Werner et al.,
2016) and iLOVECLIM (right panels, Caley et al., 2014) hydrographic data. (a) and (d) PI
Bayesian regression models between foraminifera δ18Oc and annual surface density. Data in
the PI experiments have been selected at the same locations as observations (Fig. 1).
Posterior predictive samples and the LGM δ18Oc-density relation (LGM) are visible. (b) and
(e) Density residuals (predicted - observed) for the PI experiments. (c) and (f) Density
residuals anomaly between LGM and PI. Results for the Mediterranean Sea have been
excluded because of its difficulty to be simulated and inconsistency between the two model
simulations because of their different grid resolutions. Annual mean temperature and
δ18Osw were used for the iLOVECLIM experiment whereas monthly temperature and
δ18Osw were used for the ECHAM5/MPI-OM experiment.

[Figure]

Revised Figure 6: probability distributions of surface density residuals anomaly (LGM - PI) for
ECHAM5/MPI-OM and iLOVECLIM, for global data (a and c), and without the Nordic Seas and
northern North Atlantic (north of 40°N) (b and d). North Indian Ocean data for iLOVECLIM
have been removed in both cases.

[Figure]

Figure 7 revised: reconstructions of LGM and LH annual surface ocean density from
foraminifera δ¹⁸Oc. (a) Spatial distribution of the LGM - LH absolute density anomaly. (b)
Spatial distribution of the LGM - LH density changes due to hydrographic changes in SST and
SSS. (c) Meridional gradient of reconstructed surface annual LGM density (absolute density
in dark blue, density due to hydrographic changes in light blue) and comparison with LH
reconstructions (red and orange colors). Error bars for each data point represent the 68 %
C.I. A polynomial fit (5th degree) and associated 95% confidence bands are shown as solid
resp. dashed lines. (d) Meridional gradient of reconstructed density anomaly (LGM - LH) for
absolute density in dark green and density due to hydrographic changes in light green and
associated 68 % C.I (grey lines).

**2. Uncertainty of density reconstruction.** With respect to the previous point, the uncertainty in the density reconstruction due to ocean volume changes should be implemented into the error analysis.

Based on our new analyses this uncertainty is very small (**less than 0.1 in observations and model simulations,** see discussion before and Figure b) in comparison to uncertainties in the Bayesian calibration (see Figure 2 of our paper). We performed a sensitivity test based on our LGM database for three different foraminifera species. We used a method for propagating uncertainty in density estimates through bootstrapping. It allows for the calculation of confidence intervals (CIs) that account for this additional uncertainty (estimated here to be 0.1), resulting in updated confidence intervals for the uncertainties of absolute densities (Tables 1 and 2).

| Species | Mean density (ρ) | CI Low 80% | CI High 80% | CI Low 95% | CI High 95% |
|---|---|---|---|---|---|
| N. pachyderma | 25.72 | 25.11 | 26.35 | 24.78 | 26.68 |
| G. ruber | 24.19 | 23.25 | 25.13 | 22.74 | 25.67 |
| G. bulloides | 26.18 | 25.08 | 27.31 | 24.47 | 27.90 |

Table 1: initial density prediction and uncertainties for an example of three different foraminifera species.

| Species | Mean density (ρ) | CI Low 80% | CI High 80% | CI Low 95% | CI High 95% |
|---|---|---|---|---|---|
| N. pachyderma | 25.72 | 25.09 | 26.36 | 24.75 | 26.72 |
| G. ruber | 24.19 | 23.21 | 25.16 | 22.71 | 25.69 |
| G. bulloides | 26.18 | 25.04 | 27.30 | 24.45 | 27.88 |

Table 2: density prediction and uncertainties for the same example of three different foraminifera species but adding the uncertainty in the density reconstruction due to ocean volume changes.

**Our sensitivity test of uncertainties propagation through bootstrapping indicates that the uncertainties in the density reconstruction due to ocean volume changes can be neglected.**

With respect to the uncertainty of $\delta^{18}O$ seawater reconstructions (line 72-75), I recommend citing the work of Schmidt (1999), because this author provides a reliable error estimate for $\delta^{18}O$ sea water reconstructions.

Ok, done

Also, I miss an assessment if the total error of the method is small enough to distinguish glacial densities from the modern ones, particularly if the global warming bias in the modern reference data is considered.

Uncertainties given in Figure 2 of our paper (uncertainties at 1 sigma) can be compared to changes in density in our submitted or revised Figure 7. It is dependent on the species considered but is comprised between around 0.48 and 0.86 (at 1 sigma), for reconstructed
density changes due to hydrographic changes in SST and SSS (the local/regional SST and SSS
changes mentioned by the reviewer) that vary between around 0 and 3 and absolute density
changes between around 0.8 and 3.8 (see revised Figure 7). Note that when absolute density
is considered, the signal/noise ratio increases because density increases at the LGM (by
around 0.77) but the uncertainties is similar (see our estimation discussed before). **So yes,**
**the total error of the method is small enough to distinguish absolute glacial densities from**
**the modern ones.**

Regarding a potential global warming bias in the modern reference data.

For the Multi Observation Global Ocean Sea Surface density product (Droghei et al., 2016;
2018, cmems), we used the period 1993-2003 for our calibration but the temporal extent is
only from 1993 to 2025 so we cannot investigate if older historical period can lead to
different densities compare to a period more affected by the global warming. As suggested
by the reviewer in point "**8. Global warming in modern hydrography**", There might be
products like the World Ocean Atlas that integrate over longer time periods and therefore
contain less global warming signals. We therefore investigate WOA18 observations and
compare the available periods 1995-2004 (closest period to the one we use for the
calibration) and the period 1955-1964, the oldest one proposed for WOA18 and susceptible
to be less affected by global warming effect (Figure c).

[Figure]

Figure c: difference in surface density (WOA18 1995-2004 – WOA18 1955-1964)

Results highlights no clear effect of the global warming bias, it seems more related to
uncertainties in the density product (quality and number of observations) than the global
warming bias.

At observation locations (Figure d), the effect of the global warming is very week, except
maybe in the equatorial East Atlantic but again this could be related to uncertainties in the density product rather than a global warming effect, as highlighted in Figure e for this region.

[Figure]

Figure d: difference in surface density (WOA18 1995-2004 – WOA18 1955-1964) at observation sites.

[Figure]

Figure e:  The standard deviation about the statistical mean of surface density in each grid-square (WOA18 1955-1964).

To clearly demonstrate that a potential global warming bias in the modern reference data
has no significant influence on our final density prediction, we realized a new Bayesian
calibration with WOA18 (period 1955-1964) as the modern reference data and compared it
with our previous calibration (cmems, period 1993-2003) (Figures f and g).

[Figure]

Figure f: Bayesian calibrations with (a) cmems density observations (period 1993-2003) and
(b) with WOA18 density observations (period 1955-1964).

[Figure]

Figure g: Difference in density at LGM predicted using WOA18 (period 1955-1964) observations for the calibration – predictions using cmems observations for the calibration (period 1993-2003).

We observe (Figures f and g) differences that are generally lower than 0.1 and of maximum 0.2 kg/m³. Similar results are obtained for the confidence interval calculated. As already demonstrated in response to point 2 of the reviewer, these differences can be considered as uncertainties that can be neglected.

**Therefore, the type of density product (WOA18 versus cmems observations) and the time period considered (period 1955-1964 versus 1993-2003) have no significant impact on our Bayesian calibration models and associated LGM density predictions within the estimated uncertainty.**

**3. Ice effect correction.** The authors cite an ice effect correction of either 1.0 (line 365) or 1.05 (line 421). Please clarify why different numbers have been used or correct. I also find it appropriate to cite Labeyrie et al. (1987) (see their Fig. 5) in this context, as they for the first time provided robust evidence for an ice effect on the order of 1 o/oo.

Yes, we keep 1.0 ‰ and we added references of Schrag et al., 2002 and Labeyrie et al. (1987).

**4. Transparency**. The documentation of the data sources is not sufficient. For the LGM compilation the authors mention „additional data" which are not specified anywhere in the paper. In the „Code and data availability" section, it is stated that „The additional LGM and δ18Oc dataset will be available as a supplement". Unfortunately, the supplement is not available to me. The „Obligations to authors" states that „ A paper should contain sufficient detail and references to public sources of information to permit the author's peers to replicate the work." As a reviewer, I am unable to replicate the work because neither the data/supplement nor all the sources are available to me. Also, „Copernicus Publications requests depositing data that correspond to journal articles in reliable (public) data repositories, assigning digital object identifiers, and properly citing data sets as individual contributions" (from https://www.climate-of-the-past.net/policies/data_policy.html). I strongly suggest that the authors adhere to this policy.

All the data and the code for the Bayesian calibrations are freely available at the following repository: https://github.com/nicrie/density_uncertainty. We refrained to share these data at the stage of the preprint because the data will be publicly available but still not validated and accepted for publication. We understand and agree that the reviewers should have access to these data before publication. We also shared the link with the editor.

We revised the Code and data availability section as follows: "The Python code for Bayesian calibration models is freely available at the following repository: https://github.com/nicrie/density_uncertainty. Core top data used for this analysis are from

Malevich et al. 2019 and are available at
https://agupubs.onlinelibrary.wiley.com/doi/10.1029/2019PA003576. LGM and LH δ18Oc
dataset are available at doi:10.5194/cp-10-1939-2014-supplement for Caley et al., 2014, at
https://doi.org/10.1594/PANGAEA.894229 for Waelbroeck et al., 2014 and at
https://doi.org/10.1594/PANGAEA.920596 for Tierney et al., 2020b. The additional LGM and
LH δ18Oc dataset is available at the following repository:
https://github.com/nicrie/density_uncertainty."

**5. Transport of foraminiferal shells with currents.** Currents can transport foraminiferal
shells and the isotope signals they carry over relatively large distances. Based on typical
current speeds, it can be estimated that planktic foraminifera may be transported several
degrees latitude within their lifetime. While one can argue that the effects will be minimal
because the ambient water mass is transported with the shells, discrepancies between
recorded $\delta^{18}O$ and calculated $\delta^{18}O$ may occur if foraminifera/water masses are subducted, if
water masses are mixed or in the vicinity of fronts, with the filaments from upwelling regions
or close to freshwater plumes. I suggest that the authors consider and discuss shell
transport/expatriation in addition to seasonality and vertical migration.

We agree that transport of foraminifera shells by currents is one additional process that can
potentially lead to discrepancies between recorded $\delta^{18}O$ and calculated $\delta^{18}O$ or
hydrographic data. However, as mentioned by the reviewer, one can argue that the effects
will be minimal because the ambient water mass is transported with the shells. Indeed, our
comparison between simulated $\delta^{18}Oc$ and core tops observations (Figure 4) indicates that
this effect would be weak.

In addition, this effect is implicitly included in the uncertainty. We assume here that this is a
stochastic process that will lead sometimes to a positive and sometimes to a negative error,
which accordingly inflates the credible intervals of our predictions.

We propose to add a few sentences to explain this potential additional effect in the
paragraph dealing with all potential influences on the $\delta^{18}Oc$ signal in addition to
temperature and $\delta^{18}Osw$ (lines 100-103): "Transport of foraminifera shells by currents is
another process that could lead to discrepancies between recorded $\delta^{18}Oc$ and calculated
$\delta^{18}Oc$ or hydrographic data. However, this effect is likely minimal because the ambient water
mass is transported together with the shells."

**6. Abstract.** „We developed the use of the $\delta^{18}Oc$ of planktonic foraminifera as a surface
paleodensity proxy for the whole ocean...". As the authors show obviously not for the Nordic
Seas and hence not for the global ocean surface.

Ok, we changed this for "We developed the use of the $\delta^{18}Oc$ of planktonic foraminifera as a
surface paleodensity proxy using Bayesian regression models calibrated to annual surface
density".

**7. Effect of mixing/bioturbation (line 91-95).** The paper by Köhler and Mulitza (2024) mainly deals with the detection of the carbon ion effect, not with bioturbation. Bioturbation will have a significant effect on most core tops used in this study. At typical mixed layer depths of 5-10 cm, deglacial/glacial material will be mixed with the Holocene layer below a sedimentation rate threshold of about 2 cm/kyr (see for example Broecker, 1986), mid-Holocene material (including monsoonal related salinity/density changes) at even higher sedimentation rates. For most of the MARGO core tops, there seams to be a weak stratigraphic control.

This is an issue that affects all core-top calibrations (see for example Malevich et al., 2019). However, our comparison between simulated $\delta^{18}Oc$ and core tops observations (Figure 4) indicates that this effect should be weak. If mixing/bioturbation effect was dominant, since there is no bioturbation in model simulations, we would expect no agreement on Figure 4 ($R^2 = 0.9$).

In addition, this effect is implicitly included in the uncertainty. We assume here that this is a stochastic process that will lead sometimes to a positive and sometimes to a negative error, which accordingly inflates the confidence intervals of our predictions.

Moreover, the MARGO core top data set only contains cores for which bioturbation is not severe. In low sedimentation cores, bioturbation would lead to the mixing of glacial and interglacial shells which would in turn lead to abnormally heavy Holocene $\delta^{18}Oc$. In the MARGO data set, sediment cores with sedimentation rates lower than 5 cm/ky were discarded when planktonic $\delta^{18}Oc$ values were heavier than the SST vs ($\delta^{18}Oc - \delta^{18}Ow$) regression value + 1 root mean square error (Waelbroeck et al., 2005).

We changed the text in lines 96-99 as follows: "The four processes mentioned above have not been clearly demonstrated. In addition, the carbonate ion effect has been shown to have no detectable influence (Köhler and Mulitza, 2024) and core top data have been selected to limit the bioturbation effect (Waelbroeck et al., 2005)."

**8. Global warming in modern hydrography.** The fact that all core top calibrations are affected by global warming (line 140) is not a good justification for its use. There might be products like the World Ocean Atlas that integrate over longer time periods and therefore contain less global warming signals. This issue should at least be discussed, since global ocean warming approaches the magnitude of the deglacial warming and the bias can be considerable.

This is a potential issue that affects all core-top calibrations (see for example Malevich et al., 2019, Tierney et al., 2019). But see our response to point 2. To clearly demonstrate that a potential global warming bias in the modern reference data has no influence on our final density prediction, we realized a new Bayesian calibration with WOA18 (period 1955-1964) as the modern reference data and compare with our previous calibration (cmems, period 1993-2003) (Figures f and g).

**Therefore, the type of density product (WOA18 versus cmems observations) and the time period considered (period 1955-1964 versus 1993-2003) have no significant impact on our**

**Bayesian calibration models and associated LGM density predictions within the estimated uncertainty.**

**9. Stability of the $\delta^{18}Ow$ salinity relationship.** The authors have tested the stability of the $\delta^{18}O$/salinity relationships with the results of model simulations for the LGM. I find the choice of the time slice not ideal. In the tropics and subtropics (the majority of the ocean area), the strongest precipitation changes (and hence changes in surface $\delta^{18}O$ and salinity) occur in the early to mid-Holocene with the strengthening of the Monsoon (see for example Weldeab et al. 2007). This is the time when I would expect changes in $\delta^{18}O$ of the freshwater endmember for example due to the amount effect and hence a potential instability of the $\delta^{18}O$/Salinity relation. The authors should have access to isotope-enabled model runs representing the mid-Holocene (e.g., Shi et al. 2023, co-authored by M. Werner).

Our work is really focus on the LGM and so testing the stability of the $\delta^{18}O$/salinity relationships for the MH and its potential effect on density predictions is rather out of the scope of this paper. Nonetheless, we agree that it is interesting to put into context our results regarding other climate periods. We conducted some preliminary tests using isotope-enabled model runs representing the mid-Holocene (e.g., Shi et al. 2023) in order to demonstrate that additional uncertainties due to the evolution of the $\delta^{18}O_c$-density relationship with time are globally weak and that the new calibration has great potential to be applied to other past periods and to reconstruct the past temporal evolution of ocean surface density over the quaternary.

[Figure]

Figure h: Stability of foraminifera $\delta^{18}$Oc-density relations between PI and the MH calculated
with FAME and forced by global AWI-ESM-2.1-wiso (Shi et al., 2023) hydrographic data. (a) PI
Bayesian regression models between foraminifera $\delta^{18}$Oc and annual surface density.
Posterior predictive samples and the MH $\delta^{18}$Oc-density relation (MH) are visible. (b) Density
residuals (predicted - observed) for the PI experiments. (c) Density residuals anomaly between MH and PI. (d) Probability distributions of surface density residuals anomaly (MH - PI) without Nordic Seas (north of 40°N).

Results of the Figure h clearly indicates a strong stability of foraminifera $\delta^{18}$Oc-density relations between MH and the PI and so very weak influence of $\delta^{18}$O/Salinity relation instability on final density predictions.

These results are now part of the new "Appendix B. Application of our calibration to other past periods".

**10. Direct comparison to modelled LGM density.** Why has the LGM foraminiferal-based density reconstruction not directly been compared to modelled LGM density? The models are considered good enough to test the stability of the $\delta^{18}$O salinity relation, why are they not good enough to compare with the density reconstruction directly?

This will be the focus of a detailed paper submitted very soon Barathieu et al.,. As mentioned in our paper in line 439-440: "Further regional analyses of ocean surface density and comparison with numerical climate models are presented in Barathieu et al. in prep."

We think that it would be interesting to have this detailed data-model comparison paper as a companion paper of our paper in CP if the editor agrees.

We present some results of data-model comparison from the paper of Barathieu et al., that will be submitted soon to show that the models used in our study (MPI and iLOVECLIM) exhibit significant (p-value <0.05) and strong correlation ($R^2$>0.5) with our reconstructed density for the PI and LGM (see response to reviewer 2 for details).

Broecker, W. S.: Oxygen Isotope Constraints on Surface Ocean Temperatures, Quat. res., 26, 121–134, https://doi.org/10.1016/0033-5894(86)90087-6, 1986.

Duplessy, J.-C., Labeyrie, L., Anne, Maitre, F., Duprat, J., and Sarnthein, M.: Surface salinity reconstruction of the North Atlantic Ocean during the LGM, Oceanologica Acta, 14, 311–324, 1991.

Labeyrie, L. D., Duplessy, J. C., and Blanc, P. L.: Variations in mode of formation and temperature of oceanic deep waters over the past 125,000 years, Nature, 327, 477–482, https://doi.org/10.1038/327477a0, 1987.

Köhler, P. and Mulitza, S.: No detectable influence of the carbonate ion effect on changes in stable carbon isotope ratios ($\delta^{13}$C) of shallow dwelling planktic foraminifera over the past 160 kyr, Clim. Past, 20, 991–1015, https://doi.org/10.5194/cp-20-991-2024, 2024.

Schmidt, G. A.: Error analysis of paleosalinity calculations, Paleoceanography, 14, 422–429, https://doi.org/10.1029/1999PA900008, 1999.

Shi, X., Cauquoin, A., Lohmann, G., Jonkers, L., Wang, Q., Yang, H., Sun, Y., and Werner, M.: Simulated stable water isotopes during the mid-Holocene and pre-industrial periods using AWI-ESM-2.1-wiso, Geosci. Model Dev., 16, 5153–5178, https://doi.org/10.5194/gmd-16-5153-2023, 2023.

Weldeab, S., Lea, D. W., Schneider, R. R., and Andersen, N.: 155,000 years of West African monsoon and ocean thermal evolution, Science (New York, N.Y.), 316, 1303–1307, https://doi.org/10.1126/science.1140461, 2007.

References:

Droghei, R., Nardelli, B. B., and Santoleri, R.: Combining In Situ and Satellite Observations to Retrieve Salinity and Density at the Ocean Surface, J. Atmos. Ocean. Tech., 33, 1211–1223, https://doi.org/10.1175/JTECH-D-15-0194.1, 2016.

Droghei, R., Buongiorno Nardelli, B., and Santoleri, R.: A new global sea surface salinity and density dataset from multivariate observations (1993–2016), Frontiers in Marine Science, 5, https://doi.org/10.3389/fmars.2018.00084, 2018.

Malevich, S. B., Vetter, L., and Tierney, J. E.: Global core top calibration of $\delta^{18}$O in planktic foraminifera to sea surface temperature, Paleoceanogr. Paleoclimatol., 34, 1292–1315, 2019.

Schrag, D. P., Adkins, J. F., McIntyre, K., Alexander, J. L., Hodell, D. A., Charles, C. D., & McManus, J. F. (2002). The oxygen isotopic composition of seawater during the Last Glacial Maximum. *Quaternary Science Reviews*, *21*(1-3), 331-342.

Tierney, J. E., Malevich, S. B., Gray, W., Vetter, L., & Thirumalai, K. (2019). Bayesian calibration of the Mg/Ca paleothermometer in planktic foraminifera. *Paleoceanography and Paleoclimatology*, *34*(12), 2005-2030.

Tierney, J. E., Zhu, J., King, J., Malevich, S. B., Hakim, G. J., & Poulsen, C. J. (2020). Glacial cooling and climate sensitivity revisited. *Nature*, *584*(7822), 569-573.

Waelbroeck, C., Mulitza, S., Spero, H., Dokken, T., Kiefer, T., and Cortijo, E.: A global compilation of Late Holocene planktic foraminiferal $\delta^{18}$O: Relationship between surface water temperature and $\delta^{18}$O, Quaternary Sci. Rev. 24, 853–878, 2005.

 **Response to Reviewer 2**

**General comment**

Caley et al. investigate the use of planktonic foramifera $\delta^{18}O_c$ as a surface paleodensity proxy for the whole ocean. For that, the authors applied three Bayesian regression models on $\delta^{18}O_c$ datasets to reconstruct surface paleodensity for the late Holocene (LH) and the Last Glacial Maximum (LGM). Using isotope-enabled models of different complexities, Caley et al. investigated the additional uncertainties that are introduced by the potential evolution of the $\delta^{18}O_c$-density relationship with time (i.e., from LGM to LH). Except for the Nordic Seas, the authors demonstrated that additional uncertainties are weak globally (for a LGM to LH climate change).

The objectives and the method of this study correspond to the scope of CP. The study is easy to follow, and I took pleasure to read it. I have some minor comments related to the datasets, the evaluation, and the comparison with LGM results.

**Major comments**

- The description of the LGM d18O dataset lacks details, especially, on the additional data from more recent studies (lines 130-131), which are not available with the paper (or I missed them). For the revised paper, I suggest the reviewer to provide the compilation of all the data ($\delta^{18}O$ for LH and LGM + ocean datasets) they used for this study, with the appropriate references inside.

All the data and the code for the Bayesian calibrations are freely available at the following repository: https://github.com/nicrie/density_uncertainty. We refrained to share these data at the stage of the preprint because the data will be publicly available but still not validated and accepted for publication. We understand and agree that the reviewers should have access to these data before publication. We also shared the link with the editor.

The additional dataset from more recent studies is freely available at https://github.com/nicrie/density_uncertainty but the previously published $\delta^{18}O$ data for LH and LGM are already available to download on the original repository in the "code and data availability" part: "LGM and LH $\delta^{18}O_c$ dataset are available at doi:10.5194/cp-10-1939-2014-supplement for Caley et al., 2014, at [https://doi.org/10.1594/PANGAEA.894229](https://doi.org/10.1594/PANGAEA.894229) for Waelbroeck et al., 2014 and at [https://doi.org/10.1594/PANGAEA.920596](https://doi.org/10.1594/PANGAEA.920596) for Tierney et al., 2020b."

We consider that it is important to keep the original datasets, references and citations of these previous studies as we did not change/reworked these datasets.

We revised the Code and data availability section as follows: "The Python code for Bayesian calibration models is freely available at the following repository: https://github.com/nicrie/density_uncertainty. Core top data used for this analysis are from Malevich et al. 2019 and are available at https://agupubs.onlinelibrary.wiley.com/doi/10.1029/2019PA003576. LGM and LH δ18Oc
dataset are available at doi:10.5194/cp-10-1939-2014-supplement for Caley et al., 2014, at
https://doi.org/10.1594/PANGAEA.894229 for Waelbroeck et al., 2014 and at
https://doi.org/10.1594/PANGAEA.920596 for Tierney et al., 2020b. The additional LGM and
LH δ18Oc dataset is available at the following repository:
https://github.com/nicrie/density_uncertainty."

The full density dataset re-gridded onto a common 1° × 1° spatial grid will be published and available in the Barathieu et al. paper.

- For the evaluation of the residuals under LH climate (Figure 1), why is there a like a threshold in observed data at a value of 28. Is it a problem with the data? I think it should be discussed because it gives the largest density residuals. Moreover, the authors do not discuss the strongest residuals in the Mediterranean Sea, which are probably influenced by bias in net freshwater fluxes and thermoaline circulation. The authors could use this recent study from Ayache et al. (2024, https://doi.org/10.5194/gmd-17-6627-2024).

Thank you for bringing the interesting study of Ayache et al., 2024 to our attention. The "threshold" in observed data at a value of 28 for density is because we are close to the maximum of present day's annual ocean density. In some high latitudes regions (Nordic Seas and Austral Ocean), density is already high and **temperature changes have a smaller effect**. Cold water is already dense, so cooling it further doesn't increase density as much. Consequently, we observe a **sensitivity decreases**. The rate of change of density with respect to T flattens out, meaning the system becomes less responsive (see also our response to reviewer 3). However, the density residuals are not the largest in these specific regions for the poly1_hier Bayesian model as visible on (Fig. 1f) and on the Figure (a).

We added sentences in the text to clarify this point in lines 243-252: "We observe a saturation of density values close to 28 in the calibrations that correspond to high latitudes regions (Nordic Seas and Austral Ocean). When density is already high, temperature changes have a smaller effect. Cold water is already dense, so cooling it further doesn't increase density as much. Consequently, we observe a sensitivity decrease. The rate of change of density with respect to temperature flattens out, meaning that the system becomes less responsive to temperature changes. Small changes in temperature and salinity no longer cause significant shifts in density. This behavior reflects to the non-linearity of the seawater equation of state.  Although the regression becomes less predictive in this range, the estimated density values remain correct and are not expected to change strongly as ocean surface density approaches its upper limits."

[Figure]

Figure a: Scatter plot between observed density and residuals for the poly1_hier Bayesian
model. Density in kg/m$^3$.

Regarding "the strongest residuals in the Mediterranean Sea". According to the poly1_hier
Bayesian model (Fig. 1b), there are a few points with strong residuals in its eastern part only.

We added a few sentences to explain that part of the Mediterranean Sea is characterized by
high residuals, and propose future research improvements in lines 310-319: "Strong negative
residuals are also observed in the eastern part of the Mediterranean Sea. Malevich et al.
2019 reported reduced performance of their hierarchical seasonal calibration model for
$\delta^{18}$Oc and SST in this region and attributed it to the unusual behavior of G. ruber, potentially
linked to depth-habitat migration. But estimation of seasonality for this region could also be
problematic and play a role as highlighted in the study of Ayache et al. 2024. Alternatively,
biases in Mediterranean net freshwater fluxes and thermohaline circulation could affect late
Holocene $\delta^{18}$Oc values (Ayache et al., 2014). Future modelling developments, such as the
use of high-resolution regional model in combination with the FAME module, could help to
better understand the relation between $\delta^{18}$Oc, density, temperature and $\delta^{18}$Osw during past
climate changes in the Mediterranean Sea."

• Lines 328-329: I suggest to try with the yearly $\delta^{18}$Osw values from ECHAM5/MPI-OM
to really know the effect of seasonality on calculated d18Oc. With the current
comparison, it be cannot excluded that the differences between the results using
ECHAM5/MPI-OM and iLOVECLIM is due to lower resolution of iLOVECLIM or other
missing/biased processed in this lower resolution model.

As suggested by the reviewer, we used the ECHAM5/MPI-OM yearly values of $\delta^{18}$Osw to
compute the $\delta^{18}$Oc (see Figure b) and compared the results with our Figure 4 (a) to better
assess the effect of seasonality. Results indicate a slight decrease of the R$^2$ of 0.02 and a
slight increase in RMSE of 0.06 when seasonality is not taken into account. To assess whether the difference in predictive performance between the two models was statistically
significant, a paired t-test was conducted using cross-validated $R^2$ and RMSE scores. The test
yielded a t-statistic of −5.51 and a p-value of 0.0053 for $R^2$ (and of 5.27 with a p-value of
0.0062 for RMSE), indicating a significant difference between the models. Therefore,
seasonality partly explains the weak difference between the results using ECHAM5/MPI-OM
and iLOVECLIM. Lower resolution of iLOVECLIM or other missing/biased processes in this
model could also contribute to this weak difference.

We added this in the text in lines 378-385: "We tested this hypothesis by using yearly
ECHAM5/MPI-OM values to compute the $\delta^{18}Oc$ and compared the results with those
obtained with seasonal values (shown in Figure 4a) and better assess the effect of
seasonality. Results indicate a slight decrease of the $R^2$ of 0.02 and a slight increase in RMSE
of 0.06 when seasonality is not taken into account. These differences are significant
according to paired t-tests. Therefore, seasonality partly explains the small difference
between the results using ECHAM5/MPI-OM and iLOVECLIM. Lower resolution of iLOVECLIM
or other missing/biased processes in this model could also contribute to this small
difference. »

[Figure]

Figure b: comparison between PI simulated foraminifera $\delta^{18}Oc$ (‰) (FAME module forced
with yearly ECHAM5/MPI-OM climate model hydrographic data) and observed core-top
$\delta^{18}Oc$ (‰) data. The 1:1 line is indicated.

•  For the evaluation for LGM results, the residuals in the Nordic Seas are stronger with
ECHAM5/MPI-OM than with iLOVECLIM (Figure 5). This point should be discussed
more in details by the authors.

The stronger difference in LGM – PI surface density anomaly residuals between
ECHAM5/MPI-OM and iLOVECLIM in the Nordic Seas could be explained by different simulated sea ice coverage in ECHAM5/MPI-OM vs. iLOVECLIM (Figure c). Indeed, Nordic
seas are the region with the largest difference of modeled annual SST below 0°C.

Temperature is used to calculate the $\delta^{18}Oc$ signal, ocean density and to force the FAME
module.  The PI $\delta^{18}Oc$-density relationship is used to reconstruct LGM density based on LGM
$\delta^{18}Oc$.  Then the reconstructed density is compared with modelled LGM density and so any
temperature differences in the Nordic seas (as visible in Figure c) will affect density
reconstructions and then the density residuals observed in Figure 5.

We added few sentences in the text to clarify this point in lines 460-468: "We also observe in
this region larger surface density residuals anomalies (LGM – PI) with ECHAM5/MPI-OM than
with iLOVECLIM (Figure 5c and f). This can be explained by different simulated sea ice
coverage in ECHAM5/MPI-OM compared to iLOVECLIM. Indeed, the Nordic Seas is the region
with the largest difference between the two model simulations of modeled annual SST
below 0°C (https://doi.org/10.5194/egusphere-2025-2459-AC2). Temperature is used to
calculate the $\delta^{18}Oc$ signal, ocean density and to force the FAME module. Any temperature
difference in the Nordic Seas thus affects density reconstructions and hence the density
residuals (Figure 5c and f). »

[Figure]

Figure c: Difference in LGM – PI anomaly between ECHAM5/MPI-OM and iLOVECLIM for SST.
Only region with modeled annual SST below 0°C are shown to investigate differences linked
to simulated sea ice coverage.

Moreover, I would like to see some evaluation of the reconstructed density anomalies
between LGM and LH (Figure 7a). Are there other reconstructions? Or can the authors
compare those results with modeled LGM-LH surface densities?

To our knowledge, there is no method that would provide a direct quantitative
reconstruction of ocean density at the global scale. Some methods exist based on dinocyst
assemblages, using transfer functions (see for example "Peyron, O., & Vernal, A. D. (2001).

Application of artificial neural networks (ANN) to high-latitude dinocyst assemblages for the
reconstruction of past sea-surface conditions in Arctic and sub-Arctic seas. *Journal of*
*Quaternary Science: Published for the Quaternary Research Association*, *16*(7), 699-709.")
but these reconstructions are limited to the arctic regions, where we cannot evaluate our
reconstructed densities.

This lack of method at the global scale to quantitatively reconstruct densities was one of our
main motivations to develop our work.

Our results can be compared with modeled LGM-LH surface densities and this will be the
focus of a follow-up study by Barathieu et al. that will be submitted very soon, as mentioned
in our paper in line 439-440: "Further regional analyses of ocean surface density and
comparison with numerical climate models are presented in Barathieu et al. in prep."

We think that it would be interesting to have this data-model comparison paper as a
companion paper of this paper in CP if the editor agrees.

We present some results (Figure d) of data-model comparison from the paper of Barathieu
et al., that will be submitted very soon to show that the models used in our study (MPI and
iLOVECLIM) show significant (p-value <0.05) and strong correlation ($R^2$>0.5) with our
reconstructed density for the PI and LGM.

[Figure]

Figure d: Linear regressions between absolute surface density from proxy-based
reconstructions (x-axis) and model simulations (y-axis), at the global scale. Results are shown
for the LGM period (blue) and the PI period (orange). Error bars on the x-axis represent the

95% confidence intervals of the reconstructed values. Based on Barathieu et al. in
preparation.

• I would like the authors to put into context their results regarding other climate
periods. The authors state that additional uncertainties due to the evolution of the
$\delta^{18}O_c$-density relationship with time are globally weak (lines 45-46). However, this is
true, except for the Nordic Seas, for a LGM-to-LH change. It has not been proven for
another period, such as the Last Interglacial (110-130 ka). Considering mid-Holocene
period (6 ka) raised by the reviewer #1, the changes in $\delta^{18}O$ of seawater are rather
small (+0.5‰ maximum, only, in the western Pacific Ocean according to Shi et al.,
2023 and Cauquoin et al., 2019) compared to the LGM ones.

Our work is really focus on the LGM and so testing the stability of the $\delta^{18}O$/salinity
relationships for the Last Interglacial (LIG) or MH (as asked by reviewer 1) and its potential
effect on density predictions is rather out of the scope of this paper. Nonetheless, we agree
that it is interesting to put into context our results regarding other climate periods. We
already conducted some tests using isotope-enabled model runs representing the mid-
Holocene (e.g., Shi et al. 2023) in order to demonstrate that additional uncertainties due to
the evolution of the $\delta^{18}O_c$-density relationship with time are globally weak and that the new
calibration has great potential to be applied to other past periods and to reconstruct the
past temporal evolution of ocean surface density. **Results indicate a strong stability of**
foraminifera $\delta^{18}Oc$-density relations between MH and the PI and so very weak influence of
$\delta^{18}O$/Salinity relation instability on final density predictions.

We also test the LIG time period as asked by reviewer 2. We use isotope-enabled model runs
representing the LIG at 125 kyr (e.g., corresponding to the maximum changes observed
during the LIG period according to Figure 9 of Gierz et al., 2017) (Figure e).

Again results **indicate a strong stability of** foraminifera $\delta^{18}Oc$-density relations between LIG
and the PI and so very weak influence of $\delta^{18}O$/Salinity relation instability on final density
predictions (Figure e). **This confirms and reinforces our conclusion that the new calibration**
**has great potential to be applied to other past periods and to reconstruct the past**
**temporal evolution of ocean surface density.**

The additional tests we conducted during the review process for the MH and LIG time
periods allow us to put our results regarding other climate periods into context. We
therefore included these results in a new appendix ("Appendix B. Application of our
calibration to other past periods") in the revised version, together with an explanatory text
to support and reinforce our conclusion about the new calibration and its applicability to
other past periods.

[Figure]

Figure e: Stability of foraminifera $\delta^{18}O_c$-density relations between PI and the LIG calculated
with FAME and forced by global ECHAM5/MPI-OM (Gierz et al., 2017) hydrographic data. (a)
PI Bayesian regression models between foraminifera $\delta^{18}O_c$ and annual surface density.
Posterior predictive samples and LIG $\delta^{18}O_c$-density relations (LIG) are visible. (b) Density
residuals (predicted - observed) for the PI experiments. (c) Density residuals anomaly
between LIG and PI. (d) Probability distributions of surface density residuals anomaly (LIG -
PI) without Nordic Seas (north of 40°N).

- Generally, the units are missing in the labels of figures' axes. Please check all the
  figures. Also, the panel labels are used on one complete column or row in Figures 1,
  3, 5, 6 or are absent for Figure 4. Please add a letter label for each panel in the
  figures.

Ok, we corrected this in the revised version.

**Specific comments**

- Lines 74-75: to quantify past ocean density and dynamics.

We corrected in the revised version.

- Lines 148: give the units for $\delta^{18}O_c$ (and relative to which standard) and $\rho$ (sigma-theta
  relative to a density of 1029 kg/m3?).

We changed for: "$\delta^{18}O_c$ (‰ VPDB), and annual mean surface density, $\rho$ (kg/m$^3$ relative to
water density of 1000 kg/m$^3$)"

- Section 2.4.1: specify that iLOVECLIM is an intermediate-complexity model, whereas
  ECHAM5/MPI-OM is an Earth System Model.

To be precise, we mention in the revised version "The iLOVECLIM (version 1.1.3) earth
system model of intermediate-complexity" and "use the ECHAM5/MPIOM coupled General
Circulation Model (GCM)".

- Figure 1: This is for LH period I suppose?

Yes, we revised for "Bayesian calibration models for late Holocene core-top samples against
observed density"

- Lines 235: explain a bit more that is ELPD.

The ELPD measures the expected predictive accuracy of a Bayesian model. It is defined as
the sum over all data points of the expected log posterior predictive density (see Equation
(2) in Gelman et al. (2014)). In plain words, one could say that the ELPD is the average log probability that a Bayesian model assigns to new data, summed across observations. So, in
our case, a higher ELPD means the model makes sharper and more accurate density
predictions. More details can be found in Gelman et al. (2014).

We added in the revised version in lines 189-193: « The ELPD measures the expected
predictive accuracy of a Bayesian model. It is defined as the sum over all data points of the
expected log posterior predictive density (Gelman et al., 2014). In our case, a higher ELPD
means the model makes sharper and more accurate density predictions.»

• Figure 3: give the p-values.

p-values have been added in the revised Figure 3 and revised text.

• Figure 4: Only for LH period?

We specified this in the revised version: "comparison between simulated PI foraminifera
$\delta^{18}Oc$ (FAME module forced with ECHAM5/MPI-OM and iLOVECLIM climate model
hydrographic data) and observed LH core-top $\delta^{18}Oc$ data. The 1:1 line is indicated."

• Row (a) of Figure 5: the legend for the LGM values is not clear.

We specified in the revised legend of the Figure 5: "the LGM $\delta^{18}Oc$-density relations (LGM)
are visible"

• Line 421: 1 or 1.05‰?

Yes, we keep 1.0 ‰ and we added references of Schrag et al., 2002 and Labeyrie et al.
(1987) in agreement with reviewer 1's comment.

• Lines 455-456: By applying a Bayesian regression hierarchical model to LGM and LH
$\delta^{18}O_c$ foraminifera databases, we reconstructed LGM and LH annual surface density
and found stronger LGM density…

We corrected in the revised version

**References**

Ayache, M., Dutay, J.-C., Mouchet, A., Tachikawa, K., Risi, C., and Ramstein, G.: Modelling the
water isotope distribution in the Mediterranean Sea using a high-resolution oceanic model
(NEMO-MED12-watiso v1.0): evaluation of model results against in situ observations, *Geosci.*
*Model Dev.*, **17**, 6627–6655, https://doi.org/10.5194/gmd-17-6627-2024, 2024.

Cauquoin, A., Werner, M., and Lohmann, G.: Water isotopes – climate relationships for the
mid-Holocene and preindustrial period simulated with an isotope-enabled version of MPI-
ESM, *Clim. Past*, **15**, 1913–1937, https://doi.org/10.5194/cp-15-1913-2019, 2019.

Shi, X., Cauquoin, A., Lohmann, G., Jonkers, L., Wang, Q., Yang, H., Sun, Y., and Werner, M.:
Simulated stable water isotopes during the mid-Holocene and pre-industrial periods using

AWI-ESM-2.1-wiso, *Geosci. Model Dev.*, **16**, 5153–5178, https://doi.org/10.5194/gmd-16-5153-2023, 2023.

References

Gelman, A., Hwang, J. and Vehtari, A.: Understanding predictive information criteria for Bayesian models. *Stat. Comput*., Springer US, 24, 997-1016, 2014.

Gierz, P., Werner, M., & Lohmann, G. (2017). Simulating climate and stable water isotopes during the Last Interglacial using a coupled climate-isotope model. *Journal of Advances in Modeling Earth Systems*, *9*(5), 2027-2045.

Malevich, S. B., Vetter, L., and Tierney, J. E.: Global core top calibration of $\delta^{18}O$ in planktic foraminifera to sea surface temperature, Paleoceanogr. Paleoclimatol., 34, 1292–1315, 2019.

Peyron, O., & Vernal, A. D. (2001). Application of artificial neural networks (ANN) to high-latitude dinocyst assemblages for the reconstruction of past sea-surface conditions in Arctic and sub-Arctic seas. *Journal of Quaternary Science: Published for the Quaternary Research Association*, *16*(7), 699-709.

**Response to reviewer 3**

This is a review comment on the manuscript by Caley et al submitted to Climate of The Past.

In this paper, the authors present a calibration effort of sea surface water density and d18O analyzed in mixed layer dwelling planktonic foraminifers from core top samples. This is a laudable effort as sea water density is a key parameter driving and responding to oceanographic changes. Agreeing with the comments made by the other reviewer, I will focus on some caveats of the calibration effort. These caveats limit the applicability of the calibration equation in extreme climates of the past. Hence, this limitation needs to be clear spelled out.

- It stands out that in the high salinity regions (Med Sea, Arabian Sea, Red Sea, upwelling region off NW Africa) the estimated density is less sensitive to an increase in d18Oc. This is clearly visible in Fig 1 (poly1_hier) when an increase of d18O by 3 per mill is not accompanied by a substantial predicted density change. This is issue is further highlighted in Figure 3 (lower panel), where the residual density (predicted minus observed) shows a strong correlation with salinity changes. This means that the salinity role in shaping the predicted density is underestimated.

Though less severe, this issue is also observed in low salinity regions such as the Gulf of Guinea (eastern equatorial Atlantic) and the Bay of Bengal (Northern Indian Ocean).

The implication of these observations/caveats is that the current density-d18O calibration (as presented in this paper) less reliable for the density reconstruction of past extreme climates. For instance, high d18Oc values driven large ice volume, dry climate or ocean basin characterized by anti-estuary circulation, like the current Mediterranean Sea and Red Sea). Similarly, in warming climate and wet climate (small ice sheet and large riverine runoff), this calibration is likely to provide density estimates with a large uncertainty.

While the calibration effort presents a step forward, the authors need to clearly emphasized the serious issues spelled out above. Consequently, the concluding statement made in lines 465-475 is too optimistic and needs some moderation.

We highlight the high salinity regions mentioned by reviewer 3 in the $\delta^{18}$Oc-density calibration on Figure a (with regions = Med Sea, Arabian Sea, Red Sea, upwelling off NW Africa).

[Figure]

Figure a: $\delta^{18}$Oc-density relation for late Holocene core-top samples against observed density.
We highlight in colors the high salinity regions mentioned by reviewer 3 (Med Sea, Arabian
Sea, Red Sea, upwelling region off NW Africa).

Contrary to what is stated by the reviewer 3, these regions do not correspond to the portion
of our calibration curve that is less sensitive to an increase in $\delta^{18}$Oc ("increase of d18O by 3
per mill is not accompanied by a substantial predicted density change"). In fact, except for
some parts of the Mediterranean Sea, these regions are not regions where we observe the
maximum ocean density.

In addition, we do not consider that Figure 3 shows a "strong" correlation between the
residual density and salinity changes. $R^2 = 0.2$ is a weak correlation. Also, this pooled
foraminifera species correlation integrates various species. The correlation coefficients with
SSS vary for the individual species: $R^2 = 0.17$ for *G. ruber*, $R^2 = 0.12$ for *T. sacculifer*, $R^2 = 0.54$
for *G. bulloides*, $R^2 = 0.15$ for *N. incompta*, and $R^2 = 0.32$ for *N. pachyderma* as discussed in
the text. So, probably other factors than SST and SSS influence these residual structures that
persist and some of them could indirectly be associated with gradients in SSS. For example,
negative residuals are observed in the Benguela, Canary, Peru and North Arabian regions
(Fig. 1). All these coastal areas correspond to upwelling systems and previous work already
suggested that foraminifera species could have a preference for nutrient-rich waters with
high turbidity. This is particularly true for the seasonal species *G. bulloides* (Peeters et al.,
2002; Gibson et al., 2016). The negative density residuals in these upwelling regions may
reflect this habitat preference (Fig. 1), as we discussed in the text.

The portions of the calibration curve that can be described by "when an increase of d18O by 3 per mill is not accompanied by a substantial predicted density change" correspond to some high latitude regions (Nordic Seas and Austral Ocean), as also discussed in the response to reviewer 2. This is because we are close to the maximum of density observed today.

What is the explanation of this decrease in linearity of the relation between $\delta^{18}Oc$ and surface ocean density in Nordic Seas and Austral Ocean regions? When density is already high, **temperature changes have a smaller effect**. Cold water is already dense, so cooling it further doesn't increase density as much (see TS diagram on Figure b). Consequently, we observe a **sensitivity decreases**. The rate of change of density with respect to T flattens out, meaning the system becomes less responsive to temperature changes. Small changes in temperature and salinity no longer cause significant shifts in density. This behavior is linked to the non-linearity of the seawater equation of state.

[Figure]

Figure b: TS diagram for Surface Ocean with the derivative ∂ρ/∂T in (a) and ∂ρ/∂S in (b). The derivative ∂ρ/∂T and ∂ρ/∂S represents the change in density per degree of temperature or per one salinity unit respectively. In surface waters, and at low temperatures (e.g., −2 to 2 °C), water is already dense and a temperature change has little effect: ∂ρ/∂T approaches zero. ∂ρ/∂S remains positive and relatively stable, often between 0.6 and 0.8 kg/m³ per g/kg, though it may increase slightly with salinity. Its effect becomes dominant in cold waters, where ∂ρ/∂T is weak. Both diagrams include isopycnals (lines of constant density) and have been computed with the Gibbs SeaWater (GSW) Oceanographic Toolbox of TEOS-10.

This process does not affect the $\delta^{18}Oc$ (quasi-linear fractionation with temperature at low temperature (see for example Mulitza et al., 2003)) and this is why we observe that an increase of $\delta^{18}Oc$ by 3 per mill is not accompanied by a substantial change in predicted density in Nordic Seas and Austral Ocean. Even if this part of the regression is less predictive, the estimated values of density are correct and are not expected to change strongly as ocean surface density approaches its upper limits.

In the climate model world, we found some uncertainties, in the Nordic seas, in the model simulations we conducted at the LGM because of the difficulty to simulate this region (see response to reviewer 2), together with ocean dynamic effect induced by the mean ocean salinity increase due to ice volume increase in regions of sea ice and deep water formation (see response to point 1 of reviewer 1). We therefore recommend in our paper to not apply the calibration to this region.

Regarding the fact that the calibration could be less reliable for the density reconstruction of past extreme climates. Concerning "high d18Oc values driven large ice volume", a global correction can be applied to account for the effect of ice volume increase on $\delta^{18}O$sw and salinity, and in turn on density (see our response to reviewer 1's point 1 for detailed explanations). Because it is an additional global correction, it will not change the range of values in our present day calibration.

Concerning changes between "dry and wet climates (small ice sheet and large riverine runoff)", we conducted additional test with model simulations to investigate if the calibration is likely to provide density estimates with a larger uncertainty. We agree that it is interesting to put into context our results regarding other climate periods. As asked by the reviewer 1 in point 9, we conducted some preliminary tests using isotope-enabled model runs of the mid-Holocene (e.g., Shi et al. 2023). According to reviewer 1, the strongest precipitation changes (and hence changes in surface $\delta^{18}O$ and salinity) occur in the early to mid-Holocene with the strengthening of the Monsoon. Our results clearly indicate a strong stability of foraminifera δ18Oc-density relation between the mid-Holocene (MH) and pre-industrial (PI) and thus a very weak influence of δ18O/Salinity relation instability on final density predictions. Therefore uncertainties remain within the 95% confidence interval of our calibration (see our response to point 9 of the reviewer 1). We also conducted additional tests for the last interglacial period (LIG) as requested by reviewer 2 and found a similar conclusion (see our response to reviewer 2 point "I would like the authors to put into context their results regarding other climate periods" for results). We already tested in the initial version of our paper the extreme cold and arid climate of the LGM and found that additional uncertainties are small and that our approach is valid (except for the Nordic Seas region), within propagated uncertainties from calibration into predictions of past climate conditions.

So, even if the salinity role in shaping the predicted density could be slightly underestimated (indirectly because of foraminifera ecology) for the present day calibration, applying our calibration to past extreme climates (and taking into account ecological changes) provide density predictions within the uncertainties of the calibration as demonstrated for the LGM, and now also for the MH and LIG time periods. Note that these time periods correspond to extreme climate configurations over the quaternary period as visible on Figure c, so it is reasonable to state that the new calibration has great potential to be applied to other past periods and to reconstruct the past temporal evolution of ocean surface density over the Quaternary (last 2.6 Ma).

[Figure]

Figure c: δ¹⁸O benthic foraminifera curve (LR04, Lisiecki and Raymo, 2005). Benthic δ¹⁸O
reflects ice volume and deep ocean temperature changes and is used here to highlight
extreme climatic periods (colder and more arid glacial periods versus warmer and more
humid interglacial periods). Extreme climate periods tested with isotope-enabled model runs
representing the mid-Holocene, LIG and LGM are represented by blue dots. Blue lines
indicate the range of extreme climate conditions investigated with our climate simulations
tests.

Nonetheless, we agree with reviewer 3 that under very extreme climates outside the
quaternary (see Figure c) and in ocean basin characterized by anti-estuary circulation, like
the current Mediterranean Sea and Red Sea, our calibration could provide density estimates
with larger uncertainty, a point that requires further investigations.

The additional tests we conducted during the review process for the MH and LIG time
periods allow us to put our results regarding other climate periods into context. We
therefore included these results in the new Appendix B. "Application of our calibration to
other past periods" in the revised version, together with an explanatory text to support and
reinforce our conclusion about the new calibration and its application to other past periods.

In the revised version we moderate the concluding statement made in lines 465-475 by
adding in the abstract: "The new calibration has great potential to reconstruct the past
temporal evolution of ocean surface density over the Quaternary.  Under climates outside
the Quaternary period and in ocean basins characterized by anti-estuary circulation, like the
current Mediterranean Sea and Red Sea, our calibration could provide density estimates
with larger uncertainty, a point that requires further investigations."

And we revised the conclusion as follows: "We demonstrate that our approach is valid to quantitatively reconstruct annual surface density during one of the coldest climates of the Quaternary period. We also demonstrate this for the mid Holocene and last interglacial periods (Appendix B). Hence, our calibration has great potential to be applied to other past periods and to reconstruct past temporal evolution of ocean surface density downcore during the Quaternary. Under very extreme climates outside the Quaternary (Appendix B) and in ocean basins characterized by anti-estuary circulation, like the current Mediterranean Sea and Red Sea, our calibration could provide density estimates with larger uncertainty, a point that requires further investigations."

References

Lisiecki, L. E., & Raymo, M. E. (2005). A Pliocene-Pleistocene stack of 57 globally distributed benthic δ18O records. *Paleoceanography*, *20*(1).

Mulitza, S., Boltovskoy, D., Donner, B., Meggers, H., Paul, A., & Wefer, G. (2003). Temperature: δ18O relationships of planktonic foraminifera collected from surface waters. *Palaeogeography, Palaeoclimatology, Palaeoecology*, *202*(1-2), 143-152.